

# The novel HALO mini-DOAS instrument: Inferring trace gas concentrations from air-borne UV/visible limb spectroscopy under all skies using the scaling method

Tilman Hüneke[1], Oliver-Alex Aderhold[1], Jannik Bounin[1], Marcel Dorf[1,2], Eric Gentry[1,3], Katja Grossmann[1,4], Jens-Uwe Grooß[5], Peter Hoor[6], Patrick Jöckel[7], Mareike Kenntner[1,7], Marvin Knapp[1], Matthias Knecht[1], Dominique Lörks[1], Sabrina Ludmann[1,8], Rasmus Raecke[1], Marcel Reichert[1], Jannis Weimar[1,9], Bodo Werner[1], Andreas Zahn[10], Helmut Ziereis[7], and Klaus Pfeilsticker[1]

[1]Institut für Umweltphysik, University of Heidelberg, Heidelberg, Germany
[2]Max-Planck-Institute for Chemistry, Mainz, Germany
[3]now with Department of Astronomy and Astrophysics, University of California Santa Cruz, Santa Cruz, California, USA
[4]now with Joint Institute For Regional Earth System Science and Engineering (JIFRESSE), University of California Los Angeles, Los Angeles, California, USA
[5]Forschungszentrum Jülich, Institute of Energy and Climate Research - Stratosphere (IEK-7), Jülich, Germany
[6]Institut für Physik der Atmosphäre, University of Mainz, Mainz, Germany
[7]Deutsches Zentrum für Luft- und Raumfahrt, Institut für die Physik der Atmosphäre, Oberpfaffenhofen, Germany
[8]now with Ifeu - Institut für Energie- und Umweltforschung Heidelberg GmbH, Heidelberg, Germany
[9]now with Physikalisches Institut, University of Heidelberg, Heidelberg, Germany
[10]Karlsruhe Institute of Technology (KIT), Institute for Meteorology and Climate Research, Karlsruhe, Germany

*Correspondence to:* Tilman Hüneke
(Tilman.Hueneke@iup.uni-heidelberg.de)

**Abstract.** We report on a novel 6 channel optical spectrometer (further on called mini-DOAS instrument) for aircraft-borne nadir and limb measurements of atmospheric trace gases, liquid and solid water, and spectral radiances in the UV/vis and nearIR spectral ranges. The spectrometer was developed for measurements from aboard the HALO (http://www.halo.dlr.de/) research aircraft during dedicated research missions. Here we report on the relevant instrumental details and the novel scaling

5   method used to infer the mixing ratios of UV/vis absorbing trace gases from their absorption measured in limb geometry. The uncertainties of the scaling method are assessed for $NO_2$ and BrO measurements. Some first results are reported along with complementary measurements and comparisons with model predictions for a selected HALO research flight from Cape Town to Antarctica, which was performed during the research mission ESMVal on 13 September 2012.

## 1   Introduction

10   In the past three decades aircraft-borne UV/vis spectroscopy measurements developed into a powerful tool to study the photochemistry and radiative properties of the atmosphere. Based on the pioneering work of Noxon (1975) and later Noxon et al. (1979) to exploit ground-based spectroscopic observations of the zenith scattered skylight to monitor stratospheric $NO_2$ (and later $O_3$, BrO and OClO, see below), the discovery of the ozone hole in 1985 and the need to unravel its formation mecha-



nism. Its remoteness initiated zenith sky UV/vis measurements to be not only performed from the ground (e.g., Solomon et al., 1987a) but also from research aircrafts. Accordingly, optical spectrometers were deployed on the NASA DC-8 during Airborne Arctic Stratospheric Expedition (AASE) in 1989 (e.g., Wahner et al., 1990a; Schiller et al., 1990; Wahner et al., 1990b) and later (1992 -1995) on the German Transall (e.g., Brandtjen et al., 1994; Pfeilsticker and Platt, 1994). The spectroscopic analysis

of the measured skylight spectra for the detection of $O_3$, $NO_2$, BrO, and OClO was based on Differential Optical Absorption Spectroscopy (DOAS) (Platt and Stutz, 2008), and assisting radiative transfer calculations allowed to estimate the integrated overhead (or total) column density of the targeted gases (Solomon et al., 1987b).

McElroy et al. (1999) were the first to exploit aircraft-borne nadir scattered skylight measurements to study plumes of BrO in the lower troposphere during arctic spring. Later aircraft-borne multi-axis DOAS measurements by Bruns et al.

(2004, 2006) over Europe and on major air traffic corridors by Dix et al. (2009) within the CARIBIC project (http://www. caribic-atmospheric.com/) were used to gain information on the distribution and photochemistry of pollutants and their products within the troposphere.

Meanwhile, more versatile DOAS-based 2-D imaging nadir techniques have become available to monitor the ground for sources and sinks of UV/visible/nearIR absorbing radicals, pollutants and their products and green-house gases (e.g., Heue

et al., 2008; Gerilowski et al., 2011; Merlaud et al., 2012; General et al., 2014).

Air-borne UV/vis measurements in limb geometry started with the balloon-borne study of Weidner et al. (2005) which aimed at studies of the photochemistry, budgets and trends of the $NO_x$ and $BrO_x$ families in the stratosphere (e.g., Weidner et al., 2005; Kritten et al., 2010; Kreycy et al., 2013; Kritten et al., 2014). The air-borne limb measurements of scattered skylight continued with the aircraft studies of Prados-Roman et al. (2011) made from aboard the DLR Falcon, and more recently from

the American High-performance Instrumented Airborne Platform for Environmental Research (GV HIAPER) aircraft (Baidar et al., 2013; Volkamer et al., 2015), the NSF/NCAR C-130 (Gratz et al., 2015; Ye et al., 2016), and the NASA Global Hawk (Stutz et al., 2017; Werner et al., 2017), and those reported here from the German GV aircraft HALO (for first results see Wendisch et al., 2016; Voigt et al., 2016).

One common facet of all these air-borne UV/vis limb measurements is the need for a stable observation geometry (or

pointing) of the telescopes (required are a few tenth of a degree), in order to render the underlying mathematical inversion problem for trace gas retrievals tractable (Rodgers, 2000). Therefore, all modern air-borne UV/vis spectrometers are fed by skylight collected from actively controlled telescopes to compensate for the movements (i.e., the roll and pitch angle) of the air-borne measurement platform. Most conveniently the attitude data to control the telescope's pointing are provided by the aircraft's inertial navigation system (INS) or by custom-built stabilising systems (e.g., Baidar et al., 2013).

Potentially the largest problem in the interpretation of UV/vis limb measurements involves the inherent inversion problem when assigning the observed absorption (or inferred slant column density) to the different locations in the atmosphere (Rodgers, 2000). Unfortunately, in a heavily aerosol loaded or even cloudy atmosphere, light paths (or light path distributions) are not well-defined due to multiple scattering of collected skylight. Therefore, the inversion problem becomes almost intractable when the radiative transfer forward model is not constrained by other means than the aircraft and telescope attitude, celestial,

and atmospheric pressure and temperature data. Additional data on the micro-physical properties and spatial distribution of



aerosols and cloud particles are required to constrain the inversion. These are often taken from in situ aerosol measurements, lidar or radar observations, model predictions of the spatial distribution of the measured gases, and observations or predictions of the cloud cover, et cetera. In addition, the employed retrieval strategies often rely on constraining the radiative transfer by the absorption strength of simultaneously measured absorption bands of the collisional complex $O_2-O_2$ (in the following briefly

called $O_4$) and/or relative radiances (e.g., Bruns et al., 2006; Prados-Roman et al., 2011; Baidar et al., 2013).

Constraining the radiative transfer by $O_4$ however comes with some limitations. First of all, the absorption of $O_4$ is $\propto$ $[O_2]^2$, thus skylight is much more efficiently absorbed in the lower parts of the troposphere than in the upper troposphere or stratosphere. A (a priori unknown) fraction of the UV/vis light collected in limb geometry measurements in the middle and upper troposphere or lower stratosphere may be back-scattered from lower parts of the atmosphere (Oikarinen, 2002). A

changing ground albedo or cloud cover at low levels may thus modulate (and mimic) the measured limb absorption strength of $O_4$ higher up in the atmosphere. Therefore, the scattering properties of the troposphere - even of those parts which are not being directly sampled by the telescope's field of view (FOV) - may mimic the presence (or lack) of aerosols and cloud particles at flight altitude (Stutz et al., 2017). If a significant fraction of the targeted gas is located off the telescope's field of view, assigning proper amounts of the measured gas to the correct locations in the atmosphere thus becomes ambiguous, or

even impossible. In consequence, until the recent past, the retrievals of UV/vis limb measurements had been restricted to clear or almost clear sky observations.

A third problem of air-borne DOAS measurements in the UV/vis spectral ranges addresses the need for knowing the amount of absorption of the targeted species in the background spectrum, often called Fraunhofer reference spectrum. In skylight DOAS referring all measurements to a background spectrum appears to be necessary since the measured atmospheric absorptions

are much smaller (optical densities of atmospheric absorbers typically range between $10^{-4}$ to $10^{-2}$) than those due to the Fraunhofer lines of the sun's photosphere. Different strategies are available to determine the absorption in the Fraunhofer spectrum, depending on the available observation geometries and target gas. Most easy to deal with are gases with little or negligible amounts located overhead the aircraft (e.g., $CH_2O$, $C_2H_2O_2$, HONO, often IO, OClO at daytime, et cetera ...) because their absorption in the Fraunhofer spectrum is then small or even negligible. It is far more complicated to determine

the amount of absorption in the Fraunhofer spectrum of gases with considerable (and often spatially and temporally varying) amounts located overhead the aircraft (e.g., $O_3$, $O_4$, $NO_2$, BrO, ...). Here, direct sun observations are helpful (e.g., Volkamer et al., 2015; Stutz et al., 2017), but for fast moving aircrafts the overhead column density may change too rapidly to carry out Langley-type regressions of the measured absorption as a function of air mass (cf. Gurlit et al., 2005; Dorf et al., 2008). Therefore, the amount of absorption in the Fraunhofer spectrum and its contribution to the total absorption needs to minimized

(e.g., by referring all measurements to low solar zenith angle observations at high altitude) from which the remaining absorption eventually can be calculated using model predictions (see below).

In order to render the interpretation of air-borne UV/vis limb measurements more tractable for all kind of skies, in particular for measurements in partly cloudy skies, we recently developed the so-called scaling method (Raecke, 2013; Großmann, 2014; Werner, 2015; Hüneke, 2016; Stutz et al., 2017; Werner et al., 2017). The scaling method makes use of the concentration of

a scaling gas, either in situ measured (e.g. $O_3$) or calculated (e.g. $O_4$), which is used together with the simultaneously limb

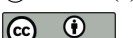



measured absorptions of the scaling gas (further on denoted $P$) and the targeted gases (further on denoted $X$), preferentially monitored in the same wavelength region. The latter appears to be convenient in order to eliminate any wavelength dependence of the atmospheric Rayleigh and Mie scattering (see Stutz et al. (2017) and their supplement, and below). The in situ measured concentration and the remotely observed absorption of the scaling gas $P$ can then be used to infer an effective light path length

(or distribution) common for the gases $P$ and $X$ (see section 3 below). The underlying assumption is a horizontally constant trace gas concentration along the line of sight equal to the in situ measured concentration. One draw-back of the scaling method comes from its (moderate) sensitivity towards the relative vertical profiles shapes (but not absolute concentrations) of the involved trace gases. The sensitivity can best be dealt with by using a scaling gas $P$ with a similar profile shape to that of the target gas $X$. The relative profile shapes of both gases can then be taken from either in situ measurements performed during

dives of the aerial vehicle, any a prior knowledge, and/or from chemistry transport models (CTMs, e.g., CLaMS, SLIMCAT) or chemistry climate models (CCMs, e.g., EMAC). The latter is very convenient since the limb measurements are often used to validate the predictions of the respective CTMs together with the other complementary measurements performed on board the respective research aircrafts.

The present study explores the scaling method in more detail together with its uncertainties and potential errors.

The paper is structured as follows. In section 2 the instrument is described and characterized. Details of the employed methods are provided in section 3. These include the spectral retrieval, radiative transfer calculation, complementary measurements, CTM and CCM modelling and a description of the scaling method and its uncertainties. Section 4 describes sensitivity studies of the retrieval method by comparing inferred $[NO_2]$ using different CTM and CCM trace gas profile predictions and different scaling gases. Finally, our results for inferred $[NO_2]$ and $[BrO]$ are inter-compared with complementary measurements and

model predictions for a HALO flight from Cape Town to Antarctica during austral spring 2012 (section 5). Section 6 concludes the study.

## 2   Instrument Description

The novel mini-DOAS instrument builds on the heritage of similar instruments assembled by our research group and collaborating partners for deployments on aircraft (e.g., the DLR Falcon, Geophysica, NASA Global Hawk, NSF/NCAR C-130) and

high flying balloons (LPMA/DOAS and MIPAS/TELIS/mini-DOAS payload) observations (Ferlemann et al., 2000; Weidner et al., 2005; Kritten et al., 2010; Prados-Roman et al., 2011; Kreycy et al., 2013; Gratz et al., 2015; Ye et al., 2016; Stutz et al., 2017; Werner et al., 2017).

Its major design criteria for air-borne measurements are a small weight (several to tens of kg), a small power consumption (200 W), multiple channels of moderate spectral resolution (i.e., ranging from several tenth of nm in UV to several nm in

nearIR) for UV/vis/nearIR analysis of the skylight received from nadir and simultaneously in scanning limb direction, a stable optical imaging, and finally an easy to operate instrument, either by on board operators (e.g., on HALO) or fully automated for deployments on unmanned aircrafts, such as the NASA Global Hawk (Stutz et al., 2017). On HALO the mini-DOAS instrument is installed in the unpressurized so-called 'boiler room' located in the rear of the HALO aircraft, which is not accessible during





the flight. While this position favors the aircraft's balance and weight distribution and provides more versatile options to assemble more maintenance-prone instruments within the cabin, it comes with the handicap of strongly changing ambient conditions to operate the instrument (i.e., boiler room temperatures may change from -30°C during polar missions to +50°C in tropical missions, and the ambient pressures may change between 1000 mbar at the ground and 150 mbar at cruise altitude),

which are prohibitive for operating stable optical instrumentation. Therefore, we follow the proven concept of our air-borne DOAS instrumentation, where the optical spectrometers are kept at vacuum pressures and temperature stabilized at 0°C by immersing the whole spectrometer container into a water-ice vessel (Weidner et al., 2005). The latter also comes with the advantages of minimising the time ($\propto$ 2h) to get the instrument flight-ready and larger auxiliary instrumentation (a cooler etc.) is not necessary in the field.

The mini-DOAS instrument consists of three major parts (Figure 1): (a) an aperture plate, from which three nadir and three limb scanning telescopes collect skylight and which is mounted into the aircraft fuselage, (b) a spectrometer unit, which houses six cooled and evacuated grating spectrometers, and (c) a control unit to automatically operate the instrument and support communication with the aircraft data network.

## 2.1 The aperture plate and telescope

The aperture plate accommodates two telescopes each for measurements in the UV, visible and nearIR spectral ranges, one of each for nadir and limb observations. It is mounted into an existing aperture opening ($28 \times 20.5 \times 9 \ \mathrm{cm}^3$) of the HALO aircraft fuselage and has a weight of 4 kg. The three limb telescopes point to the starboard side of the aircraft, perpendicular to the aircraft fuselage axis, and are moveable from $+3°$ to $-93°$, in steps of less than $0.005°$. During the flight they are commanded to compensate for the changing roll angle of the aircraft (see below), while the three nadir telescopes are held fixed. The six

telescopes have diameters of $1.2 \ \mathrm{cm}$ each, and six silica fiber bundles conduct the collected skylight from the telescopes to the spectrometers. At the spectrometer end, the fibers are linearly arranged and placed at the entrance slits of the spectrometers. At the telescope end, the fibers are linearly arranged as well positioned in the focal point of the telescope lenses, forming field of views (FOVs) of $3.15°$ in the horizontal $0.38°$ in the vertical for the UV and visible telescopes, and $1.68°$ in the horizontal and $0.76°$ in the vertical for the nearIR telescopes (for the other details see Table 1). Finally, an industrial miniature

camera is attached to the telescope aperture plate and oriented towards the sky's limb for monitoring of the investigated sky area simultaneously with the spectroscopic measurements.

## 2.2 Spectrometer unit

The six grating spectrometers are assembled in a Czerny-Turner configuration with the specifications given in Table 1. In order to clearly identify each spectrometer and the corresponding telescope, they are labeled by the wavelength range and

numbered 1 through 6. Spectrometers UV1, VIS3, and NIR5 (odd numbers) are then used in nadir viewing geometry, and spectrometers UV2, VIS4 and NIR6 (even numbers) are used in limb viewing geometry. All spectrometers are mounted onto the lid of a vacuum tight container. The spectrometer container lid also accommodates vacuum tight connectors and feed-throughs for the fiber bundles and the connection to the detector electronics. Prior to each mission the vacuum tight spectrometer





container is evacuated to some $10^{-5}$ mbar (leakage rate $2 \times 10^{-5}$ mbar $\cdot$ l/s) to keep the spectrometer and detectors clean from contamination and the optical imaging stable. The whole vacuum tight spectrometer container is immersed into a vessel filled with 7 l of water/ice, in order to stabilise the spectrometer and detector temperatures at around 0 °C. The whole spectrometer unit is further insulated using a combination of silica vacuum insulation panels (thermal conductivity of $0.008$ W/(m $\cdot$ K)) and

a more flexible Polyvenylidenfluorid (PVDF) foam (thermal conductivity of $0.037$ W/(m $\cdot$ K)). Prior to a flight, the water ice vessel is filled with approx. 4 kg of ice and 3 l of cooled water, providing a latent heat of melting of 1300 kJ. When operating under arctic conditions, i.e. with an already cooled instrument prior to flight preparations, constant temperatures are maintained for 10 hours or more, showing that average heat flows during operation are well below 36 W. In a worst case scenario, i.e. in very hot and humid ambient conditions in the tropics (e.g., in Manaus/Brazil in fall 2014, or the Maldives in August 2015), the

instrument has to be subsequently cooled adding ice and removing liquid water prior to the flight. Under these conditions, the average heat flow during flight preparation and measurement flight is around 80 W, and therefore in the present configuration the instrument is limited to $3 - 4$ hours of stable temperatures ($\Delta T \leq \pm 1$ °C). Therefore, after having made some experience with the instrument's heat budget, three Peltier elements were additionally mounted on the spectrometer container lid.

## 2.3   Control unit

The power supply, the read-out electronics for the six spectrometers, the controllers for the telescope motion, the control board for the Peltier elements, house-keeping electronics as well as a single board personal computer for instrument control and data storage and communication with the operator in the aircraft cabin is integrated into two removable electronic boxes, mounted above the spectrometer unit (yellow boxes in Figure 1). The measurements and control processes including read-out of the aircraft attitude data and the motion control of the three limb telescopes is controlled by a LabView software running on the

single board computer. Finally the whole instrument is mounted on a custom-built rack of $45 \times 47 \times 54$ cm$^3$. The total weight of the instrument is 57 kg, including the water/ice, and it consumes 100-200 W of 28 Volt DC power provided by the aircraft, depending on the power consumption of the Peltier elements.

## 2.4   Pre-flight test measurements

Prior to each mission, the instrument is optically and electronically characterized in the laboratory for a subset of parameters.

This characterization includes recording of the dark currents and offset voltage of the CCD detectors, recording of line shapes and the optical dispersion, recording of trace gas absorption spectra, measurements of the telescope's field of views, and alignment of telescopes to the major aircraft axis (roll angle).

Dark current and offset voltage: Dark current and offset voltages of the CCD detectors are recorded prior to each flight for post-flight data processing (Platt and Stutz, 2008).

Slit function: The spectrometer slit function and wavelength dispersion are monitored in the laboratory and in the field prior to each flight using HgNe and Kr emission lamps (see Table 1). Moreover, since test measurements in the laboratory show that the slit functions are sensitive to the spectrometer's temperature, their T-dependence is extensively studied and monitored in the laboratory. For example it is found that the width of the slit function is most sensitive at low temperatures with a sensitivity





of 0.005 nm/K (0.04 channels/K). However, due to the thermal stability of the instrument, a temperature sensitive slit function does not need to be taken into account for most spectral retrievals.

The effective field of view ($FOV_{eff}$) of the telescopes is made up of three contributions, which are (a) the optical FOV of the telescope ($FOV_{opt}$), (b) the lag time between aircraft movement and telescope attitude correction ($\Delta_{attit}$) and (c) the play

of the telescope gear ($\Delta_{gear}$). These are discussed in the following paragraphs.

$FOV_{opt}$ (a): The optical FOV of the telescopes is measured in the laboratory in advance of the deployment to any mission. $FOV_{opt}$ is listed in Table 1. The vertical $FOV_{opt}$ in the UV/vis is $\approx 0.38°$.

Telescope attitude control (b): In order to maintain the targeted elevation angle (EA) of the telescopes relative to the horizon during flight, the changing roll angle of the moving aircraft has to be corrected for. The aircraft's attitude data is received from

the aircraft sensor data system (BAsic HALO Measurement And Sensor system, or in brief BAHAMAS) aboard the HALO aircraft at a frequency of 10 Hz and a time delay < 1 ms via an Ethernet UDP broadcast. Due to the continuous movement of the aircraft and the time delay between data transmission and actual motor movement, a small difference between the targeted and the actual telescope angle can thus be expected. Tests involving a continuous and arbitrary sampling of the aircraft roll angle and the telescope position yields a mismatch of both angles with a standard deviation of $\Delta_{attit} \approx 0.17°...0.18°$ (Fig. 1

in the supplement).

Telescope gear (c): In addition, the pointing precision is limited by the play of the telescope's gear ($\Delta_{gear}$). Telescope gear play ($\Delta_{gear} \approx 0.05°$) is determined by the shift of the recorded radiance maximum when the telescope's FOV is measured by scanning in opposite directions.

Gaussian summation of contributions (a) ... (c) gives a $FOV_{eff}$ ranging between 0.54°(during mission ML-Cirrus (Voigt

et al., 2016)) and 0.64°(during the TACTS/ESMVal missions (e.g. Müller et al., 2016)) for the VIS4 telescope, for which the tests are carried out. Arguably it is of the same size for the other limb telescopes.

Telescope alignment to the major aircraft axis: After integration of the instrument into the aircraft, the telescope angle with respect to the aircraft is calibrated by placing a Ne gas lamp in 15 m distance and at the same height as the telescopes in the line of sight of the telescopes. The lamp is modified so that light is only emitted through a narrow ($\sim 5\,mm$) slit. Scanning over the

lamp again gives the field of view of the telescope, whose maximum is used to determine the angle that represents a horizontal line of sight with respect to the horizon. Under the assumption of a 2 cm uncertainty in the height of the lamp relative to the aperture plate (1 cm at each side), the angle uncertainty is 0.076°. When the aircraft is grounded, the aircraft roll angle given by the aircraft attitude data has a standard deviation of 0.2°. Accordingly, the systematic error in telescope alignment is $\Delta_{align}$ < 0.3°.

The systematic misalignment ($\Delta_{align}$) can be tested independently by observation of the radiance 'knee', i.e. the apparent maximum in the relative radiances received from a set of elevation angles in limb direction, which is wavelength dependent (see Figure 5 in Deutschmann et al. (2011) and Figure 5 in Weidner et al. (2005)). Figure 2 in the supplement shows measured and modelled relative radiances in the UV and visible wavelength ranges, indicating a systematic misalignment below 0.2°.





## 3 Methods

### 3.1 DOAS retrieval

The spectral retrieval is based on the DOAS method (Platt and Stutz, 2008). The primary product of the DOAS spectral retrieval are so-called differential slant column densities (dSCDs) given in molecules per $cm^2$ (Platt and Stutz, 2008), i.e., the amount

of absorption measured in a foreground versus background (Fraunhofer) spectrum. Since the details of the spectral retrieval and its uncertainties have been described in previous studies (Harder et al., 1998; Aliwell et al., 2002; Weidner et al., 2005; Dorf et al., 2006; Butz et al., 2006; Kritten et al., 2010; Stutz et al., 2017), here only those details are discussed which depart from our previous work. Table 2 provides a brief summary of the different DOAS settings and typical dSCD errors. Table 3 lists the absorption cross sections used in the analysis together with their uncertainties as stated in the literature. In all spectral

retrievals a polynomial of degree 2 is included to compensated for broad-band extinction features in the radiative transfer of the atmosphere, together with a Fraunhofer reference, a Ring spectrum and an additional Ring spectrum multiplied by $\lambda^4$ as suggested by Wagner et al. (2009). The trace gas cross section spectra are calculated by convolving the literature absorption cross sections listed in Table 3 with the measured dispersion and a Gaussian line-shape describing the Hg line at 404 nm (UV) or the Kr line at 450 nm (vis). Inaccuracies in wavelength calibration due to small changes in the instrument's optics and errors

in the wavelength calibration of the fitted spectra are accounted for during the spectral retrieval. All trace gas cross sections are linked together and the package of trace gas cross sections is allowed to shift against the Fraunhofer reference and the Ring spectra which are linked together. Typical spectral shifts for both groups of spectra are well below 1 detector pixel.

### 3.1.1 Spectral retrieval of $O_3$, BrO, OClO, $CH_2O$, and $O_4$ in the UV spectral range

Five different spectral windows are analyzed for the absorption of $O_3$, BrO, OClO, $CH_2O$, and $O_4$ in the UV wavelength

region (Tables 2 and 3). All five intervals are different but show significant overlap (Table 2).

$O_3$ is retrieved in the $335 - 362$ nm wavelength region of the Huggins band in order to achieve a larger spectral overlap with the other targeted gases in the UV spectral range which is found necessary in support of the scaling method (see section 3.6). Here $O_3$, BrO, $NO_2$, and $O_4$ references are included in the spectral retrieval (Table 2). The average error in the inferred $O_3$-dSCD is $6.4 \times 10^{16}$ molec/$cm^2$ for the UV spectral range. It is noteworthy that the spectral retrieval for $O_3$ could be

improved by using the stronger ozone absorptions bands of the Huggins band occurring towards the lower wavelength end of the UV spectrometer (310 nm), but then spectral overlap with the other gases as well as the much stronger absorption would negatively infer with the quality of the $O_3$-scaling method.

$O_4$ is retrieved in a spectral window ranging from $350 - 370$ nm in order to allow fitting of the collisional band $^1\Sigma_g^+$ + $^1\Sigma_g^+(\nu=1)$ (at 360.5 nm) (Table 2).

The BrO analysis window covers $342 - 362$ nm, the vibrational transitions $3, 0, 4, 0, 5, 0$, and $6, 0$ of the $A^2\Pi_{3/2} \longleftarrow X^2\Pi_{3/2}$ electronic transition. Reference spectra of $O_3$ for 223 K and 293 K (the latter orthogonalised to the 223 K reference spectrum) are included in the spectral retrieval together with reference spectra of $NO_2$, $CH_2O$, and $O_4$ (for the other parameters see Table 2. Figure 2 (bottom left) shows an example for the retrieval of BrO from a limb spectrum collected in the lowermost arctic





stratosphere during the Polstracc mission (http://www.polstracc.kit.edu) on January 31, 2016. Here the BrO-dSCD equals $(5.8 \pm 0.3) \times 10^{14}$ molec/cm$^2$.

OClO is retrieved in the 353 – 392 nm spectral range, i.e. of the vibrational bend and stretch transitions of the $A^2A_2 \longleftarrow X^2B_1$ electronic transition. The spectral fit includes references spectra of $O_3$ at 223 K and 293 K (the latter orthogonalised to

the 223 K reference spectrum) as well as reference spectra of $NO_2$ and $O_4$. Figure 2 (bottom right) shows an OClO retrieval from the Polstracc flight on January 31, 2016. In this case the OClO-dSCD is $(5.7 \pm 0.2) \times 10^{14}$ molec/cm$^2$.

$CH_2O$ is retrieved in a spectral window ranging from 323 to 357 nm, i.e. the rovibrational bands of the $\tilde{A}^1A_2 \longleftarrow \tilde{X}^1A_1$ electronic transitions. The spectral window is chosen in order to distinguish the signature from other trace gas absorptions in this wavelength range, particularly of $O_3$, BrO, and HONO. The spectral retrieval includes absorption cross sections of $O_3$ at

223 K and 293 K (the latter orthogonalised to the 223 K spectrum), spectra of $NO_2$ and $CH_2O$ taken at 293 K (since the bulk of $CH_2O$ is expected to be present in the lower troposphere) as well as of $O_4$, HONO, and BrO. Figure 2 (top left) shows a sample $CH_2O$ retrieval of a limb spectrum recorded during the HALO research flight above the Amazonian rain forest on Sept. 16, 2014 performed within the framework of the Acridicon mission (Wendisch et al., 2016). In this case, the $CH_2O$-dSCD amounts to $(1.28 \pm 0.05) \times 10^{17}$ molec/cm$^2$.

### 3.1.2  Spectral retrieval of $O_3$, $O_4$, $NO_2$, $H_2O$, IO, and $C_2H_2O_2$ in the visible spectral range

The main species measured in the visible spectral range are $O_3$, $O_4$, $NO_2$, and $H_2O$ and if sufficiently present IO, and $C_2H_2O_2$. Here the focus is put on the spectral retrieval of $O_3$, $O_4$, and $NO_2$, since the former two gases are used for the scaling method and the later complements the measurements of NO and total $NO_y$ by the AENEAS instrument (see section 3.4.2) on board HALO. The spectral retrieval of IO, $C_2H_2O_2$ and water vapor is not discussed further in this manuscript.

Ozone is analyzed in the 450 – 500 nm wavelength band of the Chappius absorption band. The center of both fitting window is thus shifted by 20 nm relative to $NO_2$. In the spectral retrieval, absorption cross sections of $NO_2$ at 223 K, together with $O_4$ and water vapor (Table 2) are included. The average error in the inferred $O_3$-dSCD is $4 \times 10^{17}$ molec/cm$^2$ in the visible spectral range.

The $^1\Sigma_g^+ + {}^1\Delta_g$ absorption of $O_4$ at 477.3 nm is analyzed in the 460 – 490 nm wavelength band with the same combination

of reference spectra as those used in the $O_3$ retrieval (Table 2). For $O_4$ the average retrieval error is $5.6 \times 10^{41}$ molec$^2$/cm$^5$.

$NO_2$ is thus analyzed in a relatively wide spectral window ranging from 424 – 490 nm of the sub-bands of the electronic transition $^2B_1 \longleftarrow {}^2A_1$ thus supporting both small dSCD errors while maintaining a stability of the least squares fit involved in the spectral retrieval. Reference spectra of $O_3$ at 223 K and 293 K (the latter orthogonalised to the 223 K spectrum), $O_4$ and water vapor are included in the retrieval (Table 2). Figure 2 (top right) shows an example of a spectral retrieval of $NO_2$ with

a dSCD of $(2.17 \pm 0.05) \times 10^{16}$ molec/cm$^2$ for a limb spectrum taken within the framework of the ESMVal mission close to Antarctica on 13 September 2012. The simultaneous detection of $O_3$ and $O_4$ is also evident in this spectral retrieval.



## 3.2 Determination of the amount $SCD_{ref}$

In order to obtain the total slant column density (SCD), which is needed to solve the inversion problem, the amount of absorption $SCD_{ref}$ contained in the Fraunhofer reference needs to be determined and added to the measured dSCD, i.e.

$$SCD = dSCD + SCD_{ref}, \hspace{6cm} (1)$$

where $SCD_{ref}$ is determined using (a) the so-called Langley method (i.e., a regression of dSCD as a function of total air mass), (b) inferred from a priori assumptions (for example for photo-labile species like OClO $SCD_{ref} = 0$ can be reasonably assumed for high sun), or (c) simulated if the light paths (i.e. the optical state of the atmosphere) and the concentration field of the species are well-known. As the mini-DOAS instrument is installed in the bottom of the aircraft fuselage, a direct sun light spectrum cannot be recorded, which prevents the use of method (a). Instead in most cases when methods (a) and (b)

are not feasible, $SCD_{ref}$ needs to be determined from the known RT and concentration field of the respective trace gas. For this purpose flight sections with clear sky conditions are selected and a non-linear retrieval constrained by measured relative radiances and/or $O_4$ optical densities is carried out in order to infer the aerosol extinction (e.g. Prados-Roman et al., 2011; Stutz et al., 2017). The inferred aerosol profile and the a priori trace gas concentration profiles known for example from chemical modelling (section 3.5) are then used in the RT modelling (see section 3.3) to simulate $SCD_{ref}$. In order to obtain $SCD_{ref}$ for

flights where non-linear aerosol profile retrievals are not suitable, their spectra are fitted against a reference spectrum for which $SCD_{ref}$ has previously been determined. The dSCD-offset relative to the yet-undetermined reference spectrum is then used to calculate the missing $SCD_{ref}$. The uncertainty of $SCD_{ref}$ is considerably decreased when the retrieval is referred to clear sky measurements. The largest dependencies of $SCD_{ref}$ are the aerosol optical depth at flight altitude, the trace gas concentration at flight altitude, and the overhead column. Typical $SCD_{ref}$ errors are of the order of $10 - 20\%$.

## 3.3 Radiative transfer modelling

The radiative transfer is simulated in 2D (and in selected cases in 3D, see supplement Figure 3) using version 3.5 of the Monte Carlo radiative transfer (RT) model McArtim (Deutschmann et al., 2011). The model's input is chosen according to the on board measured atmospheric temperatures and pressures, including climatological aerosol profiles from SAGE II (https://eosweb.larc.nasa.gov/project/sage2/sage2_v7_table) and Calipso (https://eosweb.larc.nasa.gov/project/calipso/cal_lid_

l3_apro_cloudfree-standard-V3-00). In the standard run, the ground albedo is set to 0.15 over sea and 0.3 over land, accounting for the surface reflectance and broken cloud cover. The RT model is further fed with the actual geolocation of the HALO aircraft, solar zenith and azimuth angles as encountered during each measurement, the telescopes azimuth and EAs, as well as the field of view (FOV) of the mini-DOAS telescopes. Stutz et al. (2017) show in their Figure 5 one example of simulated measurements for limb observations at about 18 km altitude. The simulations demonstrate that the Earth's sphericity, the correct

treatment of atmospheric refraction, cloud cover, ground albedo, celestial data, wavelength et cetera are relevant in the context of the interpretation of UV/vis/nearIR limb measurements performed within the lower and middle atmosphere (Deutschmann et al., 2011). Even though the HALO mini-DOAS spectrometers are not radiometrically calibrated on an absolute scale, past

comparison exercises with independently measured and McArtim simulated limb radiance provided confidence on the quality of the RT simulations (see Fig. 5 and Fig. 6 in Deutschmann et al. (2011), Fig. 2 in Kreycy et al. (2013), and Wolf et al. (2016)).

For the forward simulations of the trace gas absorptions measured in limb direction, the RT model is further fed with simulated trace gas curtains along the flight track (for details see section 3.5 and Figure 3, panels a and b).

## 3.4 Additional measurements

### 3.4.1 Fairo

FAIRO is a new, light-weight (14.5 kg) and accurate 2-sensor device for measuring $O_3$. It combines two techniques, i.e. (a) a UV photometer that measures the light absorption of $O_3$ at a wavelength of $\lambda = 250 - 260$ nm emitted by a UV-LED and (b) a chemiluminescence detector that monitors the chemiluminescence generated by $O_3$ on the surface of an organic dye adsorbed on dry silica gel. Both techniques are simultaneously applied in order to combine the high measurement accuracy of UV photometry with the high measurement frequency of chemiluminescence detection. The UV photometer shows a 1-$\sigma$ precision of 0.08 ppb at a measurement frequency of 0.25 Hz (and a pressure of 1 bar) and an accuracy of 1.5% (determined by the uncertainty of the $O_3$ cross section). The chemiluminescence detector shows a precision of 0.05 ppb at a measurement frequency of 12.5 Hz (Zahn et al., 2012). In post-processing the chemiluminescence detector data is calibrated using the UV photometer data. FAIRO was first deployed on HALO during the TACTS/ESMVal mission (July to September 2012); its performance was excellent during all 13 flights.

### 3.4.2 AENEAS

NO and $NO_y$ measurements on board HALO are performed using a two-channel chemiluminescence detector (AENEAS - Atmospheric nitrogen oxide measurement system) in combination with a catalytic conversion technique (Ziereis et al., 2000; Stratmann et al., 2016). A commercial two-channel chemiluminescence detector (ECO PHYSICS, Switzerland) is modified for use on board of research aircrafts. The chemiluminescence technique is widely used for the detection of atmospheric NO and relies on the emission of light in the near infrared following the reaction of NO with $O_3$ (e.g. Drummond et al., 1985). Heated gold tubes in combination with CO or $H_2$ as reducing agent are frequently used to convert all species of the odd nitrogen family ($NO_2$, $HNO_2$, $HNO_3$, $HO_2NO_2$, $N_2O_5$, PAN, ...) into NO (e.g. Bollinger et al., 1983; Fahey et al., 1985) that is subsequently detected by chemiluminescence. The conversion efficiency of the gold converter is quantified using gas phase titration of NO and $O_3$ before and after each flight with a conversion efficiency of typically more than 98%. The statistical detection limit is 7 pmol/mol for the NO measurements and 8 pmol/mol for the $NO_y$ measurements for an integration time of 1 s. The overall uncertainty for the NO and $NO_y$ measurements is 8% (6.5%) for volume mixing ratios of 0.5 nmol/mol (1 nmol/mol).

### 3.4.3 TRIHOP

The TRIHOP instrument is a three channel Quantum Cascade Laser Infrared Absorption spectrometer capable of the subsequent measurement of CO, $CO_2$, $CH_4$, and $N_2O$ (Schiller et al., 2008; Müller et al., 2016). The instrument applies Quantum



Cascade Laser Absorption Spectroscopy (QCLAS) in the mid-infrared with a multipass absorption cell (type White), which is kept at a constant pressure of p = 30 hPa and has a path length of 64 m and a volume of 2.7 L. During TACTS/ESMVal the instrument is in situ calibrated approx. every 30 min during the flights against a secondary standard of compressed ambient air. The mixing ratios of the secondary standard are determined before and after the mission in the laboratory against National

Oceanic and Atmospheric Administration (NOAA) standards. Therefore, the in-flight calibrations allow to identify and correct slow instrumental drifts in the post-flight data evaluation. The integration time for each species is 1.5 s at a duty cycle of 8 s, which finally limits the temporal resolution of the measurements. During TACTS/ESMVal TRIHOP $CH_4$, ($N_2O$) data achieved a 2-$\sigma$ precision of 10 (1.1) ppbv and stability of the instrument of 15 (2.2) ppbv, respectively, before applying the post flight data correction. The total uncertainty relative to the working standard of 18 (2.5) ppbv can be regarded as an upper limit.

## 3.5   Chemistry transport and chemistry climate models

The output of the CTM CLaMS and the CCM EMAC are used in the present study. They differ in a number of ways, in particular in their representation of dynamical features of the atmosphere and the used chemistry schemes. The models are introduced in the following and their differences are highlighted later in sections 3.7.2 and 4.2 in the context of the scaling method.

CLaMS is a Lagrangian CTM system developed at Forschungszentrum Jülich, Germany. The specific model setup is described in detail by Vogel et al. (2015). It is driven by horizontal winds from ERA-Interim reanalysis (Dee et al., 2011) provided by the European Centre for Medium-Range Weather Forecasts (ECMWF). The horizontal resolution is 100 km and the simulation period ranges from May 2012 until October 2012. It is initialized using satellite data from AURA-MLS and ACE-FTS as well as tracer-tracer-correlations. For further details of the model simulation, see Vogel et al. (2015) and refer-

ences therein. Due to its Lagrangian design, the model is especially good at representing trace gradients (e.g. the extratropical tropopause or the polar vortex edge). It should be noted that the present ClaMS simulation is not optimized in particular to reproduce photochemical processes the lower troposphere. Therefore, the employed chemistry setup does only contain reactions of importance within the stratosphere (Grooß et al., 2014) and it does neither contain sources of larger hydro-carbon compounds (e.g. VOCs and NMHCs) nor any interactions of the chemical compounds with clouds.

The ECHAM/MESSy Atmospheric Chemistry (EMAC, http://www.messy-interface.org/) model is a numerical chemistry and climate simulation system that includes sub-models describing processes in the troposphere and middle atmosphere and their interaction with oceans, land and human influences (Jöckel et al., 2010). It uses the second version of the Modular Earth Submodel System (MESSy2) to link multi-institutional computer codes. The core atmospheric model is the 5th generation European Centre Hamburg general circulation model (ECHAM5, Roeckner et al., 2006). Here, we analyse data of the RC1SD-

base-10a simulation (Jöckel et al., 2016) sampled along the aircraft flight track with the submodel S4D (Jöckel et al., 2010). The time resolution is the model time step length, i.e., 12 minutes for the applied model resolution. For the RC1SD-10a simulation, EMAC has been nudged towards ERA-Interim reanalysis data (Dee et al., 2011) to reproduce the "observed" synoptic situation in the model (for details see Jöckel et al., 2016). The model is applied in the T42L90MA-resolution, i.e.





with a spherical truncation of T42 (corresponding to a quadratic Gaussian grid of approx. 2.8 by 2.8 degrees in latitude and longitude) with 90 vertical hybrid pressure levels up to 0.01 hPa.

## 3.6 The scaling method

The scaling method makes use of the information on the relevant radiative transfer gained from a simultaneously in situ and remotely (line-of-sight) measured scaling gas $P$ and the remotely measured absorption of the target gas $X$ to infer the absolute concentration $[X]$ (Raecke, 2013; Großmann, 2014; Werner et al., 2017; Stutz et al., 2017). Ideally, the absorption bands of $X$ and $P$ (Table 2) are close to each other in order to diminish the influence of wavelength dependent Rayleigh and Mie scattering on the results. The potential advantages of the scaling method over optimal estimation come from largely removing uncertainties in radiative transfer due to aerosols and clouds.

Mathematically, the method evolves along the following lines. The total measured SCD (= dSCD + $\mathrm{SCD}_{\mathrm{ref}}$) (eq. 1) can be split into slant column densities ($[X]_i \cdot B_{X_i} \cdot z_i$) of individual atmospheric layers $i$ of thickness $z_i$ with concentrations $[X]_i$ and so called box air mass factors (BoxAMFs) $B_{X_i}$ for the targeted gas $X$ (here $\mathrm{BrO}$ and $\mathrm{NO}_2$) and the scaling gas $P$ (here $\mathrm{O}_3$ and $\mathrm{O}_4$), i.e.

$$\mathrm{SCD}_X = \sum_i [X]_i \cdot B_{X_i} \cdot z_i \tag{2}$$

$$\mathrm{SCD}_P = \sum_i [P]_i \cdot B_{P_i} \cdot z_i, \tag{3}$$

For the atmospheric layer of interest $j$, i.e. the altitude range around aircraft altitude where the limb line of sight penetrates through and most of the absorption is picked up, the concentrations for both gases can be expressed as

$$[X]_j = \frac{\mathrm{SCD}_X - \sum_{i \neq j} [X]_i \cdot B_{X_i} \cdot z_i}{B_{X_j} \cdot z_j} \tag{4}$$

$$[P]_j = \frac{\mathrm{SCD}_P - \sum_{i \neq j} [P]_i \cdot B_{P_i} \cdot z_i}{B_{P_j} \cdot z_j}. \tag{5}$$

By noting that for weak absorbers (i.e. those with optical densities much smaller than unity), the BoxAMFs $B_{X_j}$ and $B_{P_j}$ are the same for both gases $X$ and $P$ when measured in the same wavelength range, the ratio of equations 4 and 5 yields:

$$\frac{[X]_j}{[P]_j} = \left( \frac{\mathrm{SCD}_X - \sum_{i \neq j} [X]_i \cdot B_{X_i} \cdot z_i}{\mathrm{SCD}_P - \sum_{i \neq j} [P]_i \cdot B_{P_i} \cdot z_i} \right) \tag{6}$$

Further, by defining so called $\alpha$-factors ($\alpha_X$, and $\alpha_P$), which describe the fraction of the absorption in layer $j$ relative to the total atmospheric absorption for both gases, i.e.



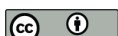

$$\alpha_{X_j} = \frac{\text{SCD}_X - \sum_{i \neq j} [X]_i \cdot B_{X_i} \cdot z_i}{\text{SCD}_X} \tag{7}$$

$$= \frac{[X]_j \cdot B_{X_j} \cdot z_j}{\sum_i [X]_i \cdot B_{X_i} \cdot z_i} \tag{8}$$

and

$$\alpha_{P_j} = \frac{\text{SCD}_P - \sum_{i \neq j} [P]_i \cdot B_{P_i} \cdot z_i}{\text{SCD}_P} \tag{9}$$

$$= \frac{[P]_j \cdot B_{P_j} \cdot z_j}{\sum_i [P]_i \cdot B_{P_i} \cdot z_i}, \tag{10}$$

the main equation of the scaling method can be written as

$$[X]_j = \frac{\alpha_{X_j}}{\alpha_{P_j}} \cdot \frac{\text{SCD}_X}{\text{SCD}_P} \cdot [P]_j \tag{11}$$

$$= \alpha_R \cdot \text{SCD}_R \cdot [P]_j. \tag{12}$$

Here $[P]_j$ is the in situ measured concentration of the scaling gas (e.g., $O_3$, $O_4$, ...), but averaged over the time of spectrum integration, and $\text{SCD}_X$, and $\text{SCD}_P$ are obtained from Eq. (1). $\alpha_R$ and $\text{SCD}_R$ are the ratios of the $\alpha$-factors and the SCDs, respectively. Equations (8) and (10) are solved using the calculated BoxAMFs $B_{X_i}$ and $B_{P_i}$ of atmospheric layer $i$ (RT model described in section 3.3) and the concentrations $[X]_i$ and $[P]_i$ from CTM/CCM predictions (section 3.5).

Figure 3 displays the major ingredients going into the scaling method. It shows CLaMS simulated curtains of concentrations of $O_3$ (panel a) and $NO_2$ (panel b), simulated BoxAMFs (panel c), and $\alpha$-factors for $O_3$ and $NO_2$ and their ratio ($\alpha_R = \frac{\alpha_{NO_2}}{\alpha_{O_3}}$) together with its uncertainty (panel d) for the HALO flight from Cape Town to Antarctica and back on 13 September 2012. Measured SCDs and their ratio are shown in panel (e) and the retrieved $NO_2$ mixing ratio in panel (f). This flight is chosen to demonstrate the key features of the method and its sensitivity to various parameters. For this flight, leading from the southern subtropics/mid-latitudes into Antarctica in spring, it is expected that (a) the overhead (stratospheric) $O_3$ and $NO_2$ concentrations largely vary in space and time and (b) at flight altitude the concentration of both gases is low and (c) in particular $NO_2$ exhibits strong concentration gradients near the tropopause and between air outside and inside the polar vortex, thus providing a critical case to test the scaling method. For this flight the RT modeled $\alpha$-factors range from 0.03 to 0.4 for $O_3$ and 0.02 to 0.3 for $NO_2$, and $\alpha_R$ ranges from 0.05 to 0.9.

Even though the $\alpha$-factors are comparably small and largely varying in space and time, the comparison of in situ measured and remotely sensed $O_3$ indicates a fairly compact relation (Figure 4). Together with RT simulations (Raecke, 2013; Knecht, 2015) this provides confidence in the retrieval of flight level trace gas concentrations from limb spectroscopy.

Evidently, the scaling and target gas are not detected at exactly the same wavelength but rather in overlapping wavelength bands. The $\lambda$-dependence of $\alpha_R$ is investigated in separate sensitivity simulations. For that purpose $\alpha$-factors are calculated for



the lower and upper wavelength end of the spectral retrieval for each gas. In agreement with Stutz et al. (2017), it is found that $\alpha_R$ may only change by as much as a few percent in our applications. Thus, the error is negligible as compared to the other errors discussed in the following section.

### 3.7 Errors of the scaling method

The errors and uncertainties of the scaling method fall into the categories random (presumably Gaussian distributed) errors and systematic errors. The sources and magnitudes of both are discussed in the following.

#### 3.7.1 Random errors of the scaling method

The random errors and sensitivities of the scaling method towards all input parameters are addressed by inspecting the Gaussian error propagation of Eq. (12). The uncertainty $\Delta[X]_j$ is calculated from

$$\Delta[X]_j = \left[ \left( \frac{\Delta \alpha_R}{\alpha_R} \right)^2 + \left( \frac{\Delta \text{SCD}_R}{\text{SCD}_R} \right)^2 + \left( \frac{\Delta[P]_j}{[P]_j} \right)^2 \right]^{0.5} \cdot [X]_j \qquad (13)$$

In the following we discuss the different contributions to $\Delta[X]_j$ in Eq. (13). The magnitudes of the contributions are summarised in Table 4.

$\Delta[P]_j$: When using in situ measured $O_3$ as scaling gas, the uncertainty $\Delta[P]_j$ is given by the uncertainty of the $O_3$ measurements (Fairo, section 3.4.1). For the comparison of in situ with limb measured $O_3$ the low frequency (0.25 Hz) precision is

obviously most relevant, since the light paths in limb direction average over extended air masses and thus in situ measured $O_3$ needs to be averaged. At 1 bar the stated $O_3$ error by Fairo is $\leq 1\%$ for $[O_3] = 40$ ppb. However, in this context more relevant are errors due to horizontal and vertical gradients in the $[O_3]$ which are considered below (see subsection (b) in the paragraph on $\Delta \alpha_R$).

When using $O_4$ as scaling gas, the altitude and temperature dependent $O_4$ concentration (in terms of $\text{molec}^2/\text{cm}^6$) can

easily be calculated with an uncertainty of $\leq \pm 1\%$ (Greenblatt et al., 1990; Pfeilsticker et al., 2001; Thalman and Volkamer, 2013).

$\left( \frac{\Delta \text{SCD}_R}{\text{SCD}_R} \right)^2 = \left( \frac{\Delta \text{SCD}_P}{\text{SCD}_P} \right)^2 + \left( \frac{\Delta \text{SCD}_X}{\text{SCD}_X} \right)^2$: The $\Delta \text{SCD}_P$ and $\Delta \text{SCD}_X$ errors each have two contributions, i.e. the dSCD errors due to the DOAS retrieval (section 3.1) and the error in determining $\text{SCD}_{\text{ref}}$ (section 3.2). The dSCD error comprises the error of the spectral retrieval and the error of the trace gas cross section. Typical dSCD errors are mentioned in section 3.1

and are often of the order of few percent. Depending on the species, the $\text{SCD}_{\text{ref}}$ errors range from 1% to 20%, but they are typically 10% (see section 3.2).

$\Delta \alpha_R$: The major contribution to the overall error $\Delta[X]_j$ may come from random errors in calculating $\alpha_R$. In the following their uncertainties (ordered into contributions (a), (b) and (c), see below) are subsequently addressed.

(a) Error due to scattering by aerosols and clouds: The influence of aerosols and clouds on $\alpha_R$ is studied from simulations

of UV/vis limb measurements in a surrogate cloud field (Figures 3 and 4 in the supplement, and Knecht (2015)). Atmospheric parameters (temperature, pressure, and cloud cover) typical for the rainy season over the Amazon (e.g. Wendisch et al. (2016)





and references therein) are assumed for the simulations, because such a scenario may represent the most severe disturbance of the radiative field in the UV/vis spectral range. The configuration of the cloud field is described in the supplement (Figure 3). For the cloudy sky, $\alpha_R$ is narrowly distributed within a range of typically $\Delta\alpha_R \leq \pm 5\%$ around the clear sky case with some outliers within an interval of $\Delta\alpha_R \leq \pm 15\%$ (Figure 4 in the supplement). A notable finding is that $\alpha_R$ follows the assumed

concentration ratio of the target gas and scaling gas, however by a somewhat damped amplitude, i.e. within an interval of $0.6 \leq \alpha_R \leq 1.8$, whereas the concentration ratio ranges between 0.2 and 1.7. In conclusion the scaling method thus largely removes the uncertainties in the concentration retrieval due the complexity of the radiative transfer in the UV/vis spectral range for a cloudy atmosphere, but the modelled $\alpha_R$ largely depend on the relative profile shapes of the target gas and scaling gas. Overall this finding is in agreement with the recent findings of Stutz et al. (2017).

(b) Uncertainties in $\alpha_R$ due to small scale variability not covered by the CTM is addressed by a comparison of CLaMS simulated and Fairo measured $O_3$ (Figure 5 in the supplement). For the HALO flight from Cape Town to Antarctica on 13 September 2012 CLaMS tends to systematically over-predict measured $O_3$ by up to 400 ppb, most likely due to errors in the vertical advection of the air masses in the sub-polar atmosphere. The impact of such a systematic error on the $O_3$ scaling is discussed below (see section 3.7.2). Moreover, the difference of measured minus simulated $[O_3]$ clusters around

several peaks with typical widths of $\Delta[O_3] \approx 40$ ppb, indicative of the sub-grid variability of $[O_3]$ not captured in the CLaMS simulations. Including the sub-grid variability in the $\alpha$-factor calculation results in $\Delta\alpha_R < 0.1$ and a typical $\Delta\alpha_R \approx 0.05$. The same comparison for the retrieved $NO_2$ results in a typical sub-grid variability of 10 ppt and a similar $\Delta\alpha_R$ as for ozone.

    (c) The telescope FOV precision and pointing accuracy (sect. 2.4) results in a rectangular window of about 500 m in height (at the location of maximum contribution to the radiance) from which the skylight is received. This is of the order of the

vertical resolution of most CTMs and CCMs. It is therefore coherent to consider an uncertainty of $\pm$ 500 m of the altitude where the vertical profile is sampled. In order to test how this uncertainty propagates into $\Delta\alpha_R$ all simulated trace gas profiles are artificially shifted by 500 m upwards and downwards and the largest and lowest $\alpha_R$ are then used as uncertainty boundaries for each measurement geometry.

    During most flight sections, $\Delta\alpha_R$ is dominated in equal parts by the uncertainty due to Mie scattering and sub-grid variabil-

ity. However, if the vertical gradient of the involved trace gases is strong around flight altitude (e.g. at 08:00 – 09:00 UTC in Figure 3), the vertical sampling uncertainty is the dominating effect (Figure 6 in the supplement). The resulting uncertainties are typically $\Delta\alpha_R \approx 10\% \dots 20\%$ for $O_3$ and $NO_2$, and in rare cases of large vertical gradients up to $\Delta\alpha_R \approx 50\%$.

### 3.7.2   Potential systematic errors of the scaling method

In our study a priori information on the profile shapes is either taken from CTM/CCM modelling, or in the case of $O_4$ from

calculations. It is thus necessary to consider how uncertainties in the predicted profile shapes propagate into the inferred concentrations at flight level.

    Systematic errors of $\alpha_R$ are investigated by modifying the involved trace gas concentration profiles in two distinct ways: By (a) changing the concentration of the scaling gas to match the in situ measured concentration while keeping the concentration of the target gas at flight altitude fixed, and (b) by shifting the CTM/CCM predicted concentration profiles of the scaling and



target gas vertically in such a way that predicted $N_2O$ concentrations at flight altitude agree with in situ measurements (Figure 7 in the supplement). It is found that errors (or biases) larger than the random error may occur if (a) the scaling gas concentration at flight altitude is significantly mispredicted by the models while the target gas concentration is not (or vice versa) or if (b) the CTM/CCM does not capture a strong vertical ascent/descent of air masses in a region with strong (and different) vertical concentration gradients of scaling and target gas. Both of these aspects need to be considered in the interpretation of measurements derived via the scaling method. For example, comparing predicted and measured concentrations of tropospheric tracers such as $CH_4$ and $N_2O$ may give confidence in the representation of ascent/descent processes near the tropopause and thus justify confidence in the predicted trace gas profile shapes.

## 4 Sensitivity studies

Sensitivity studies regarding the employed scaling gas and the employed CTM/CCM are carried out for the ESMVal flight on 13 September 2012 leading from Cape Town southwards to 65°S and back. The lower edge of the Antarctic polar vortex was penetrated during the flight between approximately 08:00 and 13:00 UTC, i.e. south of 49°S. More information on the flight, in particular the transport of dehydrated air masses form the Antarctic vortex into the upper and middle troposphere can be found in the publication of Rolf et al. (2015).

### 4.1 Intercomparison of scaling with $O_3$ and $O_4$

We compare and validate the inferred $[NO_2]$ for the HALO flight on 13 September 2012, using $O_3$ and $O_4$ as scaling gases, respectively. Figure 5 shows calculated $\alpha_R$ (panel a) and inferred $[NO_2]$ (panel b) using either $O_3$ (red symbols, further on denoted as $[NO_2]_{O_3}$) or $O_4$ (blue symbols, denoted as $[NO_2]_{O_4}$) as the scaling gas assuming clear skies (continuous lines) or a cloud layer (circles, description in the following paragraph) in the RT calculations. While the retrieved $[NO_2]_{O_3}$ and $[NO_2]_{O_4}$ agree reasonably well before 13:00 UTC, they differ after 13:00 UTC. The difference comes from the different sensitivity of $O_3$ and $O_4$ absorption towards the optical state (e.g., cloud cover) of the atmosphere. While absorption due to $O_3$ as well as $NO_2$ is largest in the stratosphere and usually smaller in the lower troposphere, it is the opposite for the absorption of $O_4$. Therefore, the shielding effect of lower and mid-level aerosols and clouds is expected to matter most for the limb detection of $O_4$ in the upper troposphere, but less for $O_3$ and $NO_2$.

The shielding effect of low and mid-level aerosols and clouds is investigated by additional RT calculations considering an uniform cloud cover (optical thickness $\tau = 20$) located at $4 - 8$ km altitude. The resulting $\alpha_R$ and inferred $[NO_2]$ are indicated as circles in panels a and b of Figure 5. Evidently including the cloud cover reduces $\alpha_R$ in $O_4$ scaling but does not significantly change $\alpha_R$ in $O_3$ scaling. Most striking is the influence of (broken) clouds on the $O_4$ scaling as evidenced by the large reduction in the calculated $\alpha_R$ for measurements prior to 8:00 UTC and after 13:00 UTC. Some proxy information on the cloud cover below the aircraft can be inferred from the colour index calculated from backscattered radiances at 600 nm / 430 nm received by the nadir VIS3 channel (panel c in Figure 5). Unlike for the time period between 9:00 and 12:30 UTC, when a more or less uniform cloud layer prevailed below the aircraft, the broken cloud cover past 13:00 UTC caused inferred $[NO_2]_{O_4}$ to become



rather variable. In contrast $[NO_2]_{O_3}$ is much less variable and closely follows the ClaMS/EMAC predicted $[NO_2]$, except for the period between 13:00 and 13:40 UTC. Here the inclusion of a cloud cover in the RT model causes $[NO_2]_{O_4}$ to converge towards $[NO_2]_{O_3}$.

Figure 6 shows the differences in inferred $[NO_2]_{O_3}$ and $[NO_2]_{O_4}$ profiles, assuming clear and cloudy skies. Evidently

inferred $[NO_2]_{O_3}$ is much less sensitive to the cloud cover than $[NO_2]_{O_4}$. The small differences (mostly $< 5\%$) at higher altitudes for inferred $[NO_2]_{O_3}$ provide confidence in the $[NO_2]_{O_3}$ retrieval for the upper troposphere and lower stratosphere. In contrast, $[NO_2]_{O_4}$ is strongly dependent on assumptions regarding the cloud cover. These results are in agreement with those reported by Stutz et al. (2017). It is worth noting that within the Antarctic troposphere $[NO_2]$ is found to be rather low ($<$ 20 ppt), and hence the systematic difference in the inferred $[NO_2]$ (up to 50% for the $[NO_2]_{O_3}$ and up to 80% for the $[NO_2]_{O_4}$)

indicate the detection limit of the DOAS limb technique for $NO_2$.

In conclusion the profile shape dependence of the scaling method thus mandates to carefully choose the scaling gas, i.e. $O_3$ appears more appropriate as a scaling gas for the detection of gases of low tropospheric and large stratospheric abundance when probed from an aircraft flying in the middle and upper troposphere and lowermost stratosphere (e.g., such as $NO_2$, $BrO$) while $O_4$ appears to be more suited for gases of large concentrations in the lower troposphere (e.g., such as $CH_2O$, $C_2H_2O_2$,

IO, and in polluted environments HONO and $NO_2$) when probed from low flying air-borne vehicles.

### 4.2   EMAC versus CLaMS profile predictions

Next, the sensitivity of inferred $[NO_2]$ as a function of predicted $O_3$ and $NO_2$ curtains is investigated. $NO_2$ mixing ratios are retrieved using trace gas curtains predicted by CLaMS (Figure 3) and EMAC (Figure 7). The retrieved $NO_2$ mixing ratios agree within the random errors during most flight sections (Figure 8, panel b). However, some differences between the models

have an impact on retrieval results, such as the higher spatial and temporal resolution of the CLaMS model. For example, a local maximum in $[NO_2]$ is predicted by CLaMS between 13:00 and 13:30 UTC but not by EMAC (Figures 3 and 7, respective panel f, and Figure 9, panel c). The retrieved $[NO_2]$ using predicted $O_3$ from CLaMS (further on denoted $[NO_2]_{O_3,CLaMS}$) is $[NO_2]_{O_3,CLaMS} \approx 0.18 \pm 0.02$ ppb, while $[NO_2]_{O_3,EMAC} \approx 0.12 \pm 0.02$ ppb. Compared with the retrieved $[NO_2]$ for this period, the CLaMS prediction appears to be overestimated, while the EMAC prediction appears to be underestimated. Thus,

model predictions with spatial resolutions comparable to the measurements (ca. 6 km horizontally) are desirable when applying the scaling method.

While good agreement is reached for $[BrO]$ in the range of $2 - 5$ ppt in the extratropical lowermost stratosphere (flight sections A and E), the difference between predicted $[BrO]_{O_3,CLaMS}$ and $[BrO]_{O_3,EMAC}$ are more substantial throughout flight sections B and D (Figure 8, panel a, and Figure 9, panel e). Two reasons for these differences can be identified. First, there is a

discrepancy in predicted tropospheric BrO concentrations between the models, which leads to a difference in calculated $\alpha_{BrO}$ at all altitudes. Below 9 km altitude, CLaMS predicts $3 - 5$ ppt, while EMAC predicts concentrations close to zero (Figure 9, panel b, dashed and dotted lines). This discrepancy is probably due to missing tropospheric sinks in the CLaMS model (sect. 3.5). Hence, the EMAC-predicted $[BrO]$ profile is expected to be more realistic. Secondly, while the extent of the polar vortex is predicted roughly in the same manner, the treatment of subsidence and methane degradation differs between the models.





This can be observed by comparing measured and predicted methane mixing ratios in flight sections B and D (Figure 8, panel c, and Figure 9, panel g). For both flight sections measurements indicate air mass ages up to 4.5 years in combination with strong dehydration (Rolf et al., 2015) and denitrification (Jurkat et al., 2017). However, the subsidence of $O_3$ appears to be overestimated in the CLaMS model, since the vertical profile of measured $O_3$ concentrations is more accurately represented

by EMAC (Figure 9, panel a).

In conclusion, differences in relative profile shapes predicted by the employed models and their spatial and temporal resolution influence the retrieval results of the scaling method. These differences are particularly large, if fundamental properties of the atmosphere, e.g. the presence of BrO in the troposphere or the subsidence in the polar vortex, are treated differently by the models. In most cases, inferred mixing ratios agree whatever model predictions (CLaMS vs. EMAC) are taken.

## 5   Sample results and discussion

Finally, we discuss the mini-DOAS observations from the flight on 13 September 2012 in the context of complementary measurements and model predictions (Figures 8 and 9). Beside the mini-DOAS measurements of $O_3$, $NO_2$, and BrO complementary instrumentation provided information on the following gases: $O_3$ from the Fairo instrument, NO and total $NO_y$ from the AENEAS instrument, and CO and $CH_4$ from the TRIHOP instrument (section 3.4). These measurements are further com-

pared with the predictions of CLaMS and EMAC, which support the interpretation with respect to the atmospheric dynamics and photochemistry. Most notable is the joint detection of NO, $NO_2$, and total $NO_y$ (and of BrO) in a remote location, such as in the Antarctic troposphere and lowermost stratosphere, since such measurements are infrequent or to date not existing. Overall, mixing ratios of BrO and $NO_2$ are inferred for the whole flight with a time resolution of  30 s and a resulting spatial resolution of  6 km, although radiative transfer implies further averaging along the line of sight (perpendicular to flight direc-

tion) of  200 km and along flight direction of  10 km. A conservative estimate for the detection limits at low mixing ratios is 2 ppt and 10 ppt for BrO and $NO_2$, respectively. Measurements of $CH_4$, which is well mixed in the troposphere and degrades in the stratosphere, provide a measure of stratospheric age of the air. Accordingly, the flight is subdivided into five flight sections A ... E (Figure 8, panel c) in order to distinguish data recorded in the midlatitude lowermost stratosphere (flight sections A and E), polar winter vortex air (flight sections B and D) and the polar troposphere (flight section C). In September 2012 the

tropospheric $CH_4$ mixing ratio at Cape Grim, Tasmania was 1778 ppb (http://www.csiro.au/greenhouse-gases/).

Inferred BrO mixing ratios are around 4 ppt / 7 ppt in flight section B and 6 ppt / 8 ppt in flight section D, based on retrievals using CLaMS / EMAC in the scaling method, respectively (Panel a of Figure 8; differences between both retrievals are discussed above in section 4.2). These concentrations are on the higher end of comparable BrO measurements in the same altitude range (12 − 13 km) reported in the literature (Harder et al., 1998; Dorf et al., 2006; Hendrick et al., 2007;

Werner et al., 2017), which could be caused by the subsidence of stratospheric air from higher altitudes discussed above. Panel e of Figure 9 shows the vertical BrO profile retrieved from the ascent of the dive at 65°S. The retrieved $[BrO]_{O_3,EMAC}$ and $[BrO]_{O_3,CLaMS}$ are both below the detection limit of 2 ppt in the altitude range below 9.5 km, even when using RT calculations based on CLaMS, which predicts 3 ppt BrO in the troposphere. Hence, below 9.5 km altitude BrO could not be detected above



the detection limit. The amount and distribution of halogen oxides such as BrO (panel e) in the troposphere is a matter of current debate (Harder et al., 1998; Fitzenberger et al., 2000; Van Roozendael et al., 2002; Saiz-Lopez and von Glasow, 2012; Volkamer et al., 2015; Wang et al., 2015; Schmidt et al., 2016; Sherwen et al., 2016; Werner et al., 2017) and is of significant scientific interest due to its potential influence on tropospheric ozone chemistry (von Glasow et al., 2004) and thus radiative

forcing (Sherwen et al., 2017). Reported tropospheric background profiles at polar latitudes include those by Fitzenberger et al. (2000), who derive tropospheric BrO profiles above Kiruna (Sweden) from balloon measurements and conclude that tropospheric [BrO] amounting to $0.4 - 2.3$ ppt eventually was present, assuming a uniform distribution within the troposphere. Prados-Roman et al. (2011) use air-borne DOAS measurements based in Spitzbergen to derive a BrO mixing ratio profile in Arctic spring with 15 ppt in the planetary boundary layer (PBL), 1.5 ppt in the free troposphere (FT), and up to 6 ppt at 10

km in the lowermost stratosphere (LMS). The measurements derived in the present study are compatible with these previously inferred background profiles and do not show elevated BrO concentrations in the Antarctic spring in September 2012.

   Retrieved $NO_2$ (Figure 8, panel b) exhibits similar features as the independently measured NO. In polar vortex air (flight sections B and D), $[NO_2]$ is mostly between 5 and 20 ppt, i.e. near or below the detection limit of 10 ppt, similar to the in situ measured [NO]. Such small amounts of $NO_x$ limit the deactivation of active chlorine, i.e. the formation of $ClONO_2$, and

thus prolong ozone destruction in the polar winter vortex air. Interestingly, enhanced $NO_2$ together with increased NO are detected in the free troposphere ($9 - 13$ km altitude) during the dive (Figure 9, panels b and c). The largest [NO] of $60 - 80$ ppt is measured around $10 - 12$ km, while the largest $[NO_2]$ of $30 - 40$ ppt is inferred around $11 - 13$ km. Increased $NO_2$ concentrations are also predicted by the CLaMS model (Figure 3, panel b), indicating in-mixing of tropospheric air from more $NO_x$ rich latitudes. Rolf et al. (2015) also infer in-mixing of moister mid-latitude air into the bottom of the polar vortex, albeit

not at the same time during the flight, since the GLORIA instrument was switched off during the dive. At altitudes below 9 and 10 km, respectively, retrieved [NO] and $[NO_2]$ are near their detection limits of 7 ppt and 10 ppt, indicating very pristine air.

## 6   Conclusions

We describe a novel six channel optical spectrometer for aircraft-borne limb and nadir detection of UV/visible/nearIR absorbing gases (e.g., $O_3$, $O_4$, $NO_2$, HONO, $CH_2O$, $C_2H_2O_2$, BrO, IO, OClO, and others), liquid and solid water as well as of skylight

radiances. Further, features of a novel retrieval method (called scaling method) are discussed which go beyond those recently reported by Stutz et al. (2017) and Werner et al. (2017). Here we demonstrate how absolute concentrations of the UV/vis absorbing gases can be inferred from limb measurements in the troposphere and lower stratosphere under all (clear and cloudy) skies. The novel scaling methods largely avoids ambiguities in the necessary mathematical inversion to interpret the limb measurements which are introduced by the complexity of radiative transfer in the UV/vis spectral range for cloudy and heavily

aerosol-loaded atmospheres. Instead, the relative profile shapes of the scaling gas and the target gas are the main a priori information used in the scaling method. Thus, uncertainties in the trace gas retrieval are primarily due to uncertainties in the relative profile shapes, which can be minimized when the retrieval uses a priori profile shapes for example from CTM/CCM predictions, calculations (e.g. $O_4$), or otherwise available measurements. The present study examines the resulting random and





systematic errors of trace gas concentrations retrieved via the scaling method. The random error is estimated to be 10 – 20 % for most measurement conditions, dominated in equal parts by uncertainties due to Mie extinction and small scale variabilities of the concentrations of the involved trace gases. The random error is comparatively large close to strong vertical or horizontal trace gas concentration gradients. Systematic biases can occur when trace gas profile shapes are strongly misrepresented by

model predictions. Thus, comparing independent trace gas measurements of e.g. tropospheric or stratospheric tracers with model predictions is essential in the interpretation of retrieval results. For limb measurements in the upper troposphere and lower stratosphere the comparison of both scaling gases indicates a sensitivity of $O_4$ scaling for low clouds, while the $O_3$ scaling is insensitive. This is consistent with the expectation that a scaling gas with similar profile shape as compared to the target gas is best suited for the method. The comparison of retrievals involving a CTM (CLaMS) and a CCM (EMAC) reveals

that results are in agreement within the random error, as long as the fundamental properties of the atmosphere are represented in a similar way (e.g. presence or absence of a trace gas in the troposphere). Further, the comparison indicates that CTM/CCM curtains with spatial resolutions close to those of the measurements are desirable.

The present study shows the applicability of the scaling method to HALO mini-DOAS measurements of $NO_2$ and $BrO$ at altitudes between 3.5 and 15 km under all sky conditions. It can be argued that the scaling method replaces the major a

priori used in the traditional optimal estimation (i.e. the aerosol and cloud profile) by a different a priori (i.e. the relative trace gas profile shape). The latter a priori has the advantages that (a) it is more homogeneous in space and time on the scales relevant for the air-borne DOAS measurements, and (b) it can be predicted more reliably by modern CTMs/CCMs as compared to the presence of aerosols and clouds. Thus, the scaling method provides a novel and reliable means for inferring trace gas concentrations from air-borne UV/vis limb measurements. The significantly decreased dependency on aerosol and

cloud properties increases the ability to make use of already recorded data and decidedly widens the applicability of air-borne UV/vis limb spectroscopy as a means of investigating atmospheric photochemistry.

*Acknowledgements.* This study was funded through the Deutsche ForschungsGemeinschaft, DFG (grants PF-384/7-1, PF-384/7-2, PF384/9-1, PF384/9-2, and PF-384/16-1). Additional funding from EU-SHIVA (FP7-ENV-2007-1-226224) is highly acknowledged. The authors gratefully acknowledge the computing time granted on the supercomputer JURECA at Jülich Supercomputing Centre (JSC) under the VSR

project ID JICG11. The EMAC simulations have been performed at the German Climate Computing Centre (DKRZ) through support from the Bundesministerium für Bildung und Forschung (BMBF). DKRZ and its scientific steering committee are gratefully acknowledged for providing the HPC and data archiving resources for this consortial project ESCiMo (Earth System Chemistry integrated Modelling). We thank the Deutsches Zentrum für Luft- und Raumfahrt (DLR) for the support to get the instrument certificated and the DLR Flugexperimente Team at Oberpfaffenhofen, in particular Heinrich Brockstieger, Frank Probst, Martina Hierle, and Andrea Hausold, for the support given

during the TACTS/ESMVal, NarVal, Cirrus, Acridicon, OMO, and Polstracc missions.





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



**Table 1.** Optical specification of the mini-DOAS instrument

| Channel name | UV | VIS | NIR |
|---|---|---|---|
| Telescope focal length and f-number | 30 mm, f/2.5 | | |
| Telescope lense coating | UV-AR | VIS 0° | NIR II |
| Telescope Schott filter type | BG3 | GG400 | RG850 |
| Number of fibers and diameter | $7 \times 200\ \mu$m | | $2 \times 400\ \mu$m |
| Fibre bundle entrance slit dimension | $1652\ \mu$m $\times\ 200\ \mu$m | | $884\ \mu$m $\times\ 400\ \mu$m |
| FOV$_{\mathrm{opt}}$ $2 \cdot \gamma$ | $3.15° \times 0.38°$ | | $1.68° \times 0.76°$ |
| Spectrometer entrance slit dimension | $1500\ \mu$m $\times\ 100\ \mu$m | | $500\ \mu$m $\times\ 100\ \mu$m |
| Spectrometer focal length and f-number | 60 mm, f/4 | | |
| Grating [grooves/mm] | 2100 | 1300 | 300 |
| Sensor name | S10141-1107S | | G9204-512 |
| Sensor type | Si-CCD | | InGaAs-PDA |
| Number of channels on sensor | 2048 | | 512 |
| Sensor area per channel | $12 \times 1464\ \mu$m$^2$ | | $15 \times 500\ \mu$m$^2$ |
| Full Well Capacity | $2 \cdot 10^5\ e^-$ | | $1.87 \cdot 10^8\ e^-$ |
| Quantum Efficiency (i) | 0.60 | 0.85 | 0.80 |
| Covered wavelength range | 310 - 440 nm | 420 - 640 nm | 1100 - 1680 nm |
| Resolution (slit function FWHM) | 0.47 nm / 6.1 px | 1.1 nm / 8.4 px | $\approx 10$ nm / $\approx 11$ px |

(i) corresponding to the wavelengths of 400 nm (UV), 540 nm (VIS) bzw. 1500 nm (NIR).



**Table 2.** Details of the spectral analysis of various trace gases

| Target gas | $\lambda$ (nm) | Fitted absorbers | Add. Param. | Polyn. | $\sigma$(dSCD) |
|---|---|---|---|---|---|
| O$_4$ | 350 – 370 | 1, 2, 3, 5, 7, 9 | $I_{\text{Ofs}}$(i), $R$ (ii), $R \cdot \lambda^4$ | 2 | |
| | 460 – 490 | 1, 2, 4, 6 | $I_{\text{Ofs}}, R, R \cdot \lambda^4$ | 2 | $5 \times 10^{41}$ |
| O$_3$ | 335 – 362 | 1, 2, 4, 7, 9 | $I_{\text{Ofs}}, R, R \cdot \lambda^4$ | 2 | |
| | 450 – 500 | 1, 2, 4, 6 | $I_{\text{Ofs}}, R, R \cdot \lambda^4$ | 2 | $4 \times 10^{18}$ |
| NO$_2$ | 407 – 435 | 1, 2, 3, 4, 6, 10 | $I_{\text{Ofs}}, R, R \cdot \lambda^4$ | 2 | |
| | 424 – 490 | 1, 2/3, 4/5, 6 | $I_{\text{Ofs}}, R, R \cdot \lambda^4$ | 2 | $2 \times 10^{15}$ |
| H$_2$O | 490 – 520 | 1, 2, 5, 6 | $I_{\text{Ofs}}, R, R \cdot \lambda^4$ | 2 | |
| HCHO | 323 – 357 | 1, 2, 3, 5, 7, 8, 9 | $I_{\text{Ofs}}, R, R \cdot \lambda^4$ | 2 | $7 \times 10^{15}$ |
| HONO | 337 – 372 | 1, 2, 3, 4, 7, 8, 9 | $I_{\text{Ofs}}, R, R \cdot \lambda^4$ | 2 | |
| BrO | 342 – 363 | 1, 2, 3, 4, 7, 9 | $I_{\text{Ofs}}, R, R \cdot \lambda^4$ | 2 | $2 \times 10^{13}$ |
| OClO | 353 – 392 | 1, 2, 3, 4, 10 | $I_{\text{Ofs}}, R, R \cdot \lambda^4$ | 2 | $3 \times 10^{13}$ |

(i) $I_{\text{Ofs}}$: Offset spectrum; (ii) R: Ring spectrum; (iii) $R \cdot \lambda^4$: Ring spectrum multiplied with $\lambda^4$.





**Table 3.** Trace gas absorptions cross sections used for the DOAS retrieval.

| No. | Absorber | Temp. | Reference | Uncertainty |
|-----|----------|-------|-----------|-------------|
| 1 | $O_4$ | 293 K | Thalman and Volkamer (2013) | 4% |
| 2 | $O_3$ | 223 K | Gorshelev et al. (2014); Serdyuchenko et al. (2014) | 3% |
| 3 | $O_3$ | 293 K | Gorshelev et al. (2014); Serdyuchenko et al. (2014) | 3% |
| 4 | $NO_2$ | 223 K | Bogumil et al. (2003) | 3.4% |
| 5 | $NO_2$ | 293 K | Bogumil et al. (2003) | 3.4% |
| 6 | $H_2O$ | 273 K | Rothman et al. (2009) | |
| 7 | HCHO | 293 K | Chance and Orphal (2011) | 5% |
| 8 | HONO | 298 K | Stutz et al. (2000) | 5% |
| 9 | BrO | 223 K | Fleischmann et al. (2004) | 10% |
| 10 | OClO | 213 K | Kromminga et al. (2003) | 5% |





**Table 4.** Summary of random errors as discussed in section 3.7.1. The percentages in columns three and four refer to deviations of the parameter in the first column.

| Parameter | Cause of the error | Typical value | Maximum value |
|---|---|---|---|
| $\Delta\alpha_R$ | RTM noise | 3.5% | 3.5% |
| | Mie scattering | 10% | 15% |
| | small scale variability | $0 - 20\%$ | 100% |
| | vertical sampling | $0 - 10\%$ | 60% |
| $\Delta\mathrm{SCD}_R$ | DOAS fit error | 5% | 100% |
| | cross section | 3% | 6% |
| | $\mathrm{SCD}_{\mathrm{ref}}$ | $5 - 10\%$ | 20% |
| $\Delta[X]$ | $O_3$ measurement | <1% | 1% |
| | $O_4$ calculation | 1% | 1% |


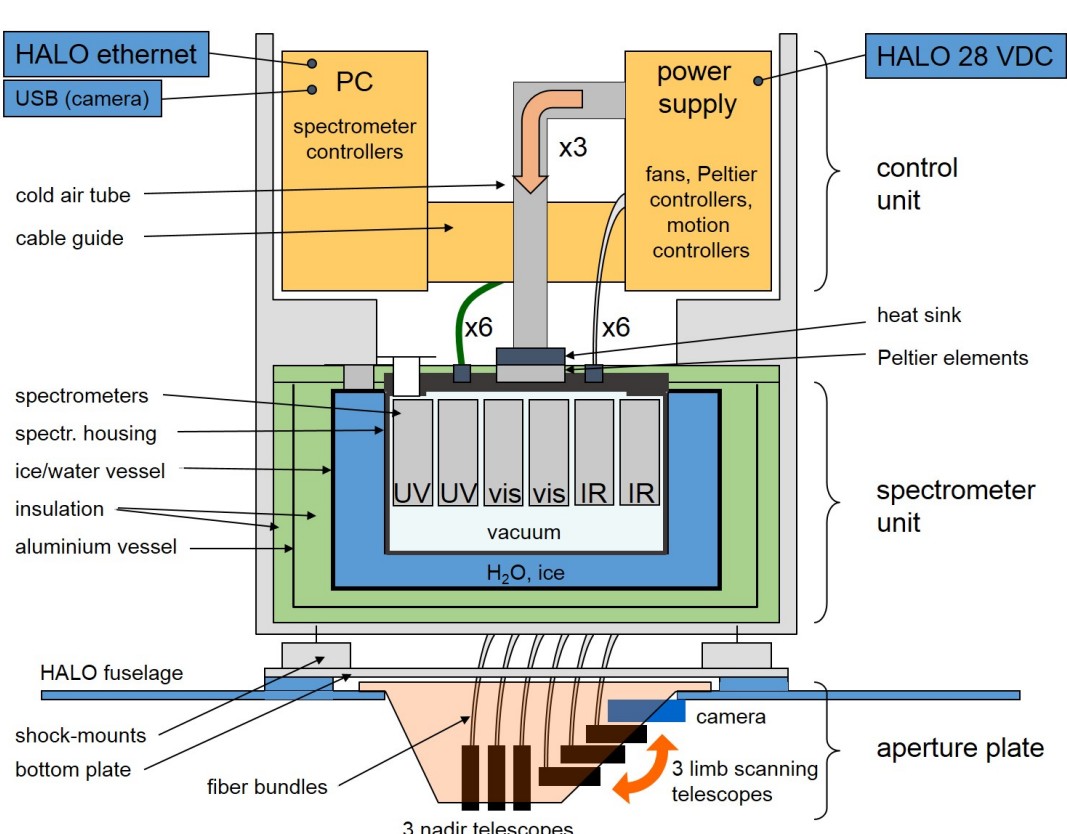

**Figure 1.** Sketch of the HALO mini-DOAS instrument.





**Figure 2.** Sample spectral retrievals as described in section 3.1. Tob left: CH₂O retrieval in the UV spectral range. Top right: NO₂ retrieval in the visible spectral range. Bottom left: BrO retrieval in the UV. Bottom right: OClO in the UV. Fitted reference absorption cross sections are shown in red and the residual structures are shown in blue.





**Figure 3.** Illustration of $NO_2$ mixing ratio retrieval for the ESMVal flight on 13 September 2012 using the CTM CLaMS. Panel a: CLaMS predicted $[O_3]$ curtain (colour scale $\times$ $7.9 \cdot 10^{12}$ cm$^{-3}$) and aircraft altitude (red line). Panel b: CLaMS predicted $[NO_2]$ curtain (colour scale $\times$ $2.9 \cdot 10^{9}$ cm$^{-3}$) and aircraft altitude (red line). Panel c: BoxAMFs calculated by the RTM McArtim (colour scale $\times \log(217)$). Panel d: Calculated $\alpha_{O_3}$ (blue) and $\alpha_{NO_2}$ (red) as well as $\alpha_R$ (black line) and its uncertainty range (grey shaded area). Panel e: Retrieved $SCD_{O_3}$ (blue, scale divided by $9.0 \times 10^{20}$) and $SCD_{NO_2}$ (red, scale divided by $3.4 \times 10^{17}$) as well as $SCD_R$ (black line, scale multiplied by $10^4$) and its uncertainty range (grey shaded area). Panel f: Retrieved $[NO_2]_{O_3}$ (light red line) and its uncertainty range (grey shaded area) together with in situ measured $O_3$ (blue line). The dark red line shows the $NO_2$ mixing ratios predicted by the CLaMS model.



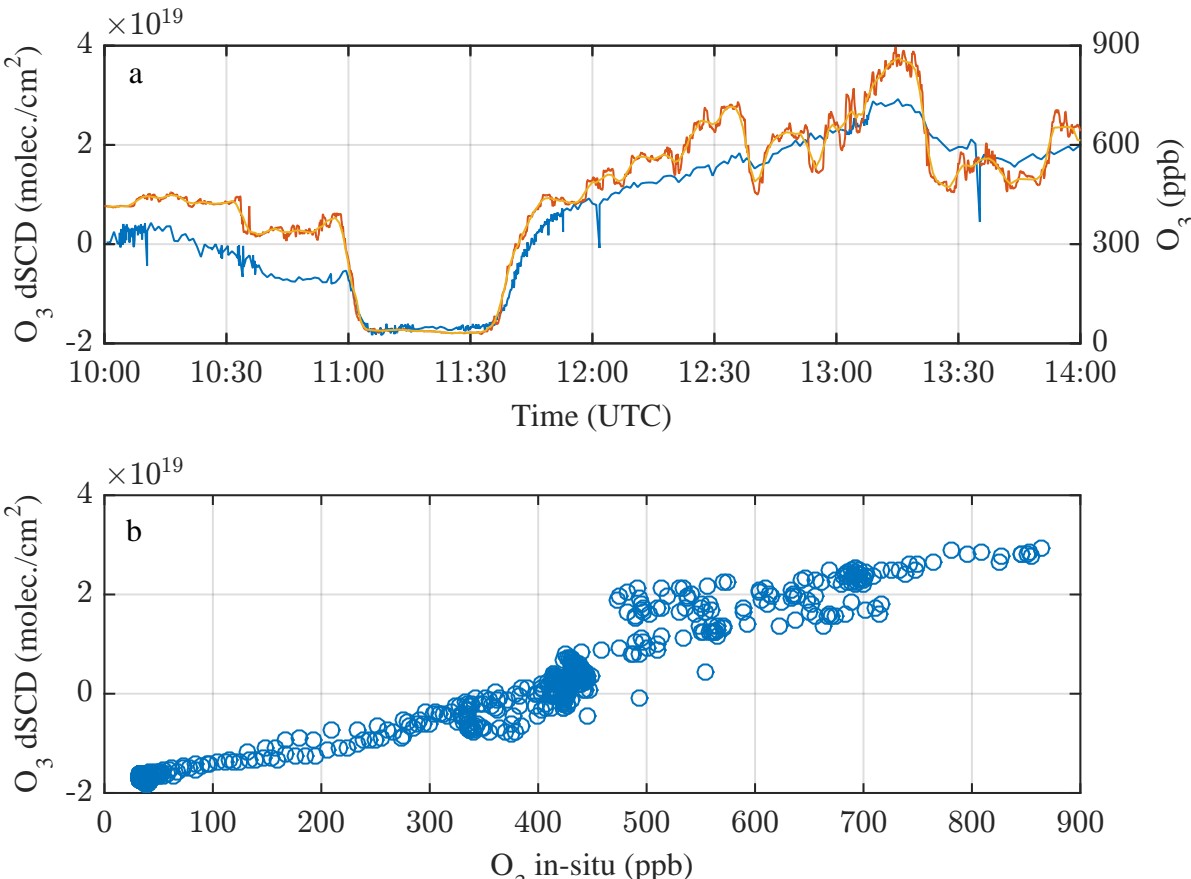

**Figure 4.** Comparison of in situ measured and remotely sensed $O_3$. Panel a: Time series of high time resolution (red line) and five minutes running average (orange line) of $O_3$ measured by the Fairo instrument, and remotely sensed $O_3$ (blue line) for a segment of the HALO flight from Cape Town to Antarctica on 13 September 2012. Panel b: Scatterplot of averaged in situ measurements and remotely sensed $O_3$ for the flight segment shown in panel a.



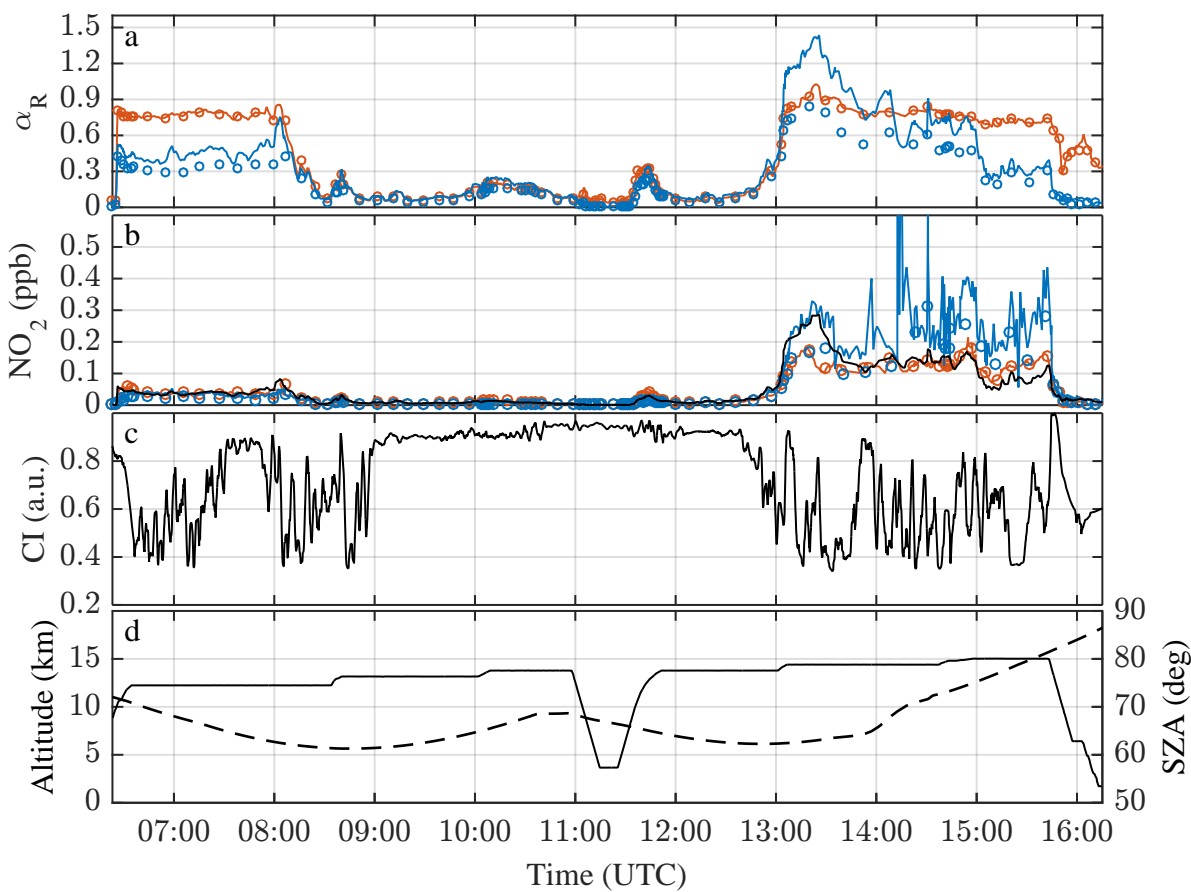

**Figure 5.** Retrieved $[NO_2]_{O_3}$ (red) and $[NO_2]_{O_4}$ (blue) for the ESMVal research flight on 13 September 2012. Calculations assuming clear skies are displayed as lines, calculations including a cloud layer at $4 - 8$ km are displayed as circles. Panel a: Timeseries of calculated $\alpha_R$. Panel b: Timeseries of inferred $[NO_2]$ together with $NO_2$ concentrations as predicted by CLaMs (black line). Panel c: Colour index (CI, 600 nm / 430 nm radiances) observed by the VIS3 channel in nadir geometry. A large/small colour index indicates a cloud cover/clear sky below the aircraft, respectively. Panel d: Pressure altitude of HALO (black line) and solar zenith angle (SZA, black dashed line).





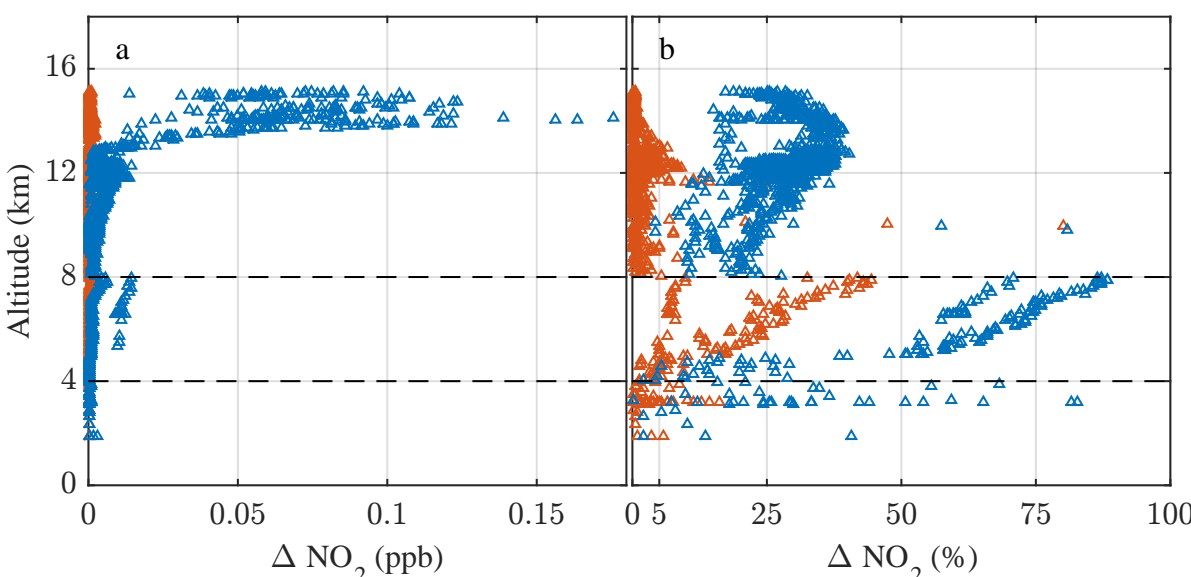

**Figure 6.** Impact of a cloud layer on retrieved $[NO_2]_{O_3}$ (red) and $[NO_2]_{O_4}$ (blue) for the ESMVal research flight on 13 September 2012. Shown are altitude profiles of the difference $\Delta[NO_2] = |[NO_2]_{clear} - [NO_2]_{clouded}|$ of the clear sky and clouded sky calculations, calculated from the data shown in Figure 5, panel b. The altitude range of the cloud layer as encountered during the dive and implemented in the clouded sky calculations is indicated by dashed lines.





**Figure 7.** Illustration of $NO_2$ mixing ratio retrieval for the ESMVal flight on 13 September 2012 using the CCM EMAC. Panel a: EMAC predicted $[O_3]$ curtain (colour scale $\times 7.9 \cdot 10^{12}$ cm$^{-3}$) and aircraft altitude (red line). Panel b: EMAC predicted $NO_2$ curtain (colour scale $\times 2.9 \cdot 10^9$ cm$^{-3}$) and aircraft altitude (red line). Panel c: Box-AMFs calculated by the RTM McArtim (colour scale $\times \log(217)$). Panel d: Calculated $\alpha_{O_3}$ (blue) and $\alpha_{NO_2}$ (red) as well as $\alpha_R$ (black line) and its uncertainty range (grey shaded area). Panel e: Retrieved $SCD_{O_3}$ (blue, scale divided by $9.0 \times 10^{20}$) and $SCD_{NO_2}$ (red, scale divided by $3.4 \times 10^{17}$) as well as $SCD_R$ (black line, scale multiplied by $10^4$) and its uncertainty range (grey shaded area). Panel f: Retrieved $[NO_2]_{O_3}$ (light red line) and its uncertainty range (grey shaded area) together with in situ measured $O_3$ (blue line). The dark red line shows the $NO_2$ mixing ratio predicted by the EMAC model.




**Figure 8.** Time series of measured trace gas mixing ratios recorded during the ESMVal research flight on 13 September 2012. Panel a: In situ measured concentration of $O_3$ (grey) and inferred $[BrO]_{O_3}$ (green) using profile shape predictions by ClaMS (red) and EMAC (green). Panel b: In situ measured NO (grey) and inferred $[NO_2]_{O_3}$ using profile shape predictions by CLaMS (red) and EMAC (green). The uncertainties are discussed in section 3.7.1. Panel c: Pressure altitude of HALO (black) and $CH_4$ mixing ratios (blue), the latter as derived from in situ measurements by TRIHOP (continuous line), CLaMS prediction (dashed line), EMAC prediction (dotted line). Additionally, flight sections A through E are marked for reference in the text.




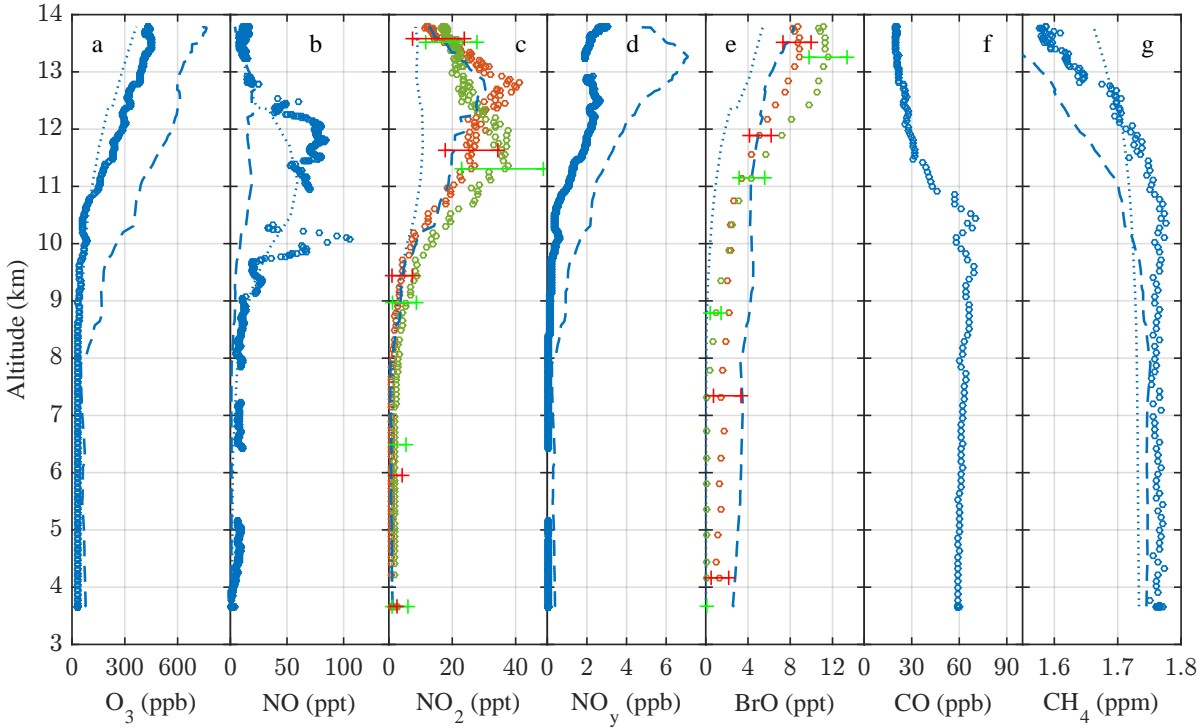

**Figure 9.** Altitude profiles of trace gas concentrations measured during the ascent (65°S, 21°E) of flight section C of the ESMVal research flight on 13 September 2012. Panel a: Measurements of $O_3$ (Fairo), panel b: NO (AENEAS), panel c: $NO_2$ (mini-DOAS), $NO_y$ (AENEAS, panel d), BrO(mini-DOAS instrument, panel e), CO (TRIHOP, panel f), and $CH_4$ (TRIHOP, panel g) are shown. Panels a, b, d, f, g: In situ measurements are indicated as blue circles, CLaMS predictions as dashed lines, and EMAC preidictions as dotted lines. Panels c, e: Inferred $[NO_2]_{O_3}$ and $[BrO]_{O_3}$ (circles), respectively, with random errors (error bars) using profile shapes predicted by CLaMS (red symbols) and EMAC (green symbols).