# Peer review of "The novel HALO mini-DOAS instrument: Inferring trace gas concentrations from air-borne UV/visible limb spectroscopy under all skies using the scaling method"

_Atmospheric Measurement Techniques, 2017_

## Referee Comment (RC1) · Anonymous Referee #1 · 20 Jun 2017

The manuscript describes the mini-DOAS instrument that has already been operated in a number of science campaigns onboard the German High Altitude and Long-range research aircraft (HALO) together with the so-called "scalling method" to infer trace gas mixing ratios from the mini-DOAS limb measurements. The paper is generally well written and provides an important reference for the mini-DOAS instrument and in particular its data analysis. I recommend publication in AMT after consideration of the following – mostly minor – comments.

Specific comments

As the instrument has already been flown on a number of science campaigns and this paper gives only a few selected examples, I suggest to include a table listing deloyments of the mini-DOAS instrument on HALO campaigns so far.

p1.l3: please spell out HALO as "German High Altitude and Long-range research aircraft (HALO)" when used for the first time and move the URL reference from the abstract to the introduction section.

p2.l1: sentence ends early

p2.l6: better either give an earlier reference for DOAS as well, or explicitly indicate that Platt and Stutz (2008) is a recent review of DOAS and not the original reference

p2.l22: "German GV" -> "German High Altitude and Long Range Research Aircraft (HALO), that is based on a Gulfstream G550 jet"

p2.l35: what exactly is meant by "celestial" here?

p3.l29: "to" -> "to be"

p4.l11: what exactly does "latter" here refer to? Only CCMs or CTMs and CCMs?

p5.l18: insert "elevation angles"

p5.l20: why "skylight" and not simply "light"'? I know its picky, but in general there may be other light sources than just skylight.

p6.l9: Can you explain briefly why surface temperatures are important here and not the much cooler upper tropospheric / lower stratospheric temperatures at cruise altitude?

p7.l10: any reference to BAHAMAS ?

p9.l1: "Polstracc" -> "POLSTRACC"

P9.l14: "Acridicon" -> "ACRIDICON"

p10.l27: What is "EA"? Elevation angle?

p11.l6: "Fairo" -> "FAIRO"

p11.l7: "for measuring" -> "for in-situ measuring"

p11.l16: not sure if "all 13 flights" is still relevant here, as many more flights on more recent campaign have been performed with FAIRO

p12.l17: horizontal resolution is not well defined for a Lagrangian model. Please give more detail what this refers to.

p12/13: EMAC: Maybe include a few sentences about EMAC's chemistry scheme - in contrast or comparison to what has been mentioned for CLaMS.

p14.l12: Suggestion: Say again explicitly that SCD_X and SCD_P are measured by the mini-DOAS, the alpha are from a model and [P] is measured in-situ. (If that is what you are doing.)

p20.l11: better be a bit more specific of where no elevated BrO concentrations are observed. I assume this statement does not refer to the Antarctic boundary layer?

Caption Fig. 2: Suggestion: Include in the caption date, location and campaign of sample measurement

---

## Referee Comment (RC2) · Anonymous Referee #2 · 22 Aug 2017

The authors describe an airborne remote sensing instrument for UV/vis/near IR scattered sunlight measurements and the method for retrieving trace gas concentrations for the flight altitude, namely DOAS and the scaling method. Further the uncertainties are described to some extent, and some first application together with a comparison with auxiliary in situ measurements are presented.

I see two main problems with the manuscript and the scaling method: a) the uncertainty on the retrieved trace gas concentrations associated with the assumptions on the profile shape in the calculation of the scaling factor appears to be the dominant uncertainty and it is poorly quantified (partially hidden away in the supplementary and mentioned in section 4.2) and b) information on the radiative transfer in the atmosphere can only be obtained from layers where the scaling gas is actually present (This is hinted at in section 4.1). The authors claim that the scaling method is superior to optimal estimation. However, the advantage of optimal estimation is that it provides a formal framework for assessing the information content of a retrieved quantity. I believe the paper would be much stronger if these issues would be properly discussed upfront. Hence I recommend publication only after major revisions in sections 3.6, 3.7, 4, and 6.

**General comments:**

P. 13-14: I'm not sure why the authors need such a lengthy mathematical description. In principle it is an intercept theorem:

$$\frac{[X]_j^{theo}}{[P]_j^{theo}} \frac{SCD_P^{theo}}{SCD_X^{theo}} = \frac{[X]_j}{[P]_j} \frac{SCD_P}{SCD_X}$$

Intuitively, this is easier to understand.

P. 16, l. 28 – P. 17, l. 8: I would suggest using the uncertainty of the model/climatological profiles and do a proper error propagation to estimate the uncertainty. The NO2 mixing ratio is up to four times larger than the standard case at one point in the probed flight. Is that realistic?

P. 18, l. 6-9: This shouldn't be just discussed as differences between different model parametrisations. Basically what it says it that you need good prior knowledge in order to retrieve a meaningful concentration with the scaling method.

P. 3, l. 33-34: The scaling method can hardly be described as novel method when the authors themselves already quote 6 reference for it.

P.1, l. 4-5: Since the paper is about the scaling method, maybe add a bit more explanation in the abstract what is actually done.

P. 21, l. 15-21: The a priori for optimal estimation is also a trace gas profile. Aerosol and cloud profiles are auxiliary information. The a priori for the scaling method is the profile ratio and aerosols and clouds are still auxiliary information. You have shown that those can be neglected. However, this is just how RT works for airborne measurements and hence this statement is valid for both techniques.

The authors poorly describe the need for the scaling method in the introduction. They list 3 main problems with airborne DOAS measurements on page 3. However, the first two problems are somewhat

convoluted and boil down to the remote sensing technique being ill-constrained. The third problem is about the residual trace gas amount in the background measurement. This seems somewhat inflated in its description (also in section 3.2 where the applied technique is then described). The authors then don't make use of their own arguments to describe the scaling method for providing a different a priori information in comparison to what has been done before.

The authors are not very precise in their referencing (see examples below). Please check carefully throughout the manuscript.

**Specific comments:**

P.1, l. 11-13: not a complete sentence

P.2, l. 1-2: 'It's remoteness initiated…'. I feel the authors are overstating here. Another reason for aircraft measurements would be to study the full extent of the ozone hole and not only locally. Please clarify or add reference for original statement.

P. 2, l. 8: 'aircraft-borne' sounds clunky.

P. 2, l. 11: How do you obtain information of 'photochemistry of pollutants' from column measurements? Please elaborate.

P. 2, l. 13-15: '… monitor the ground for sources and sinks…': Of the cited references only General et al. (2014) describe an anti-correlation between BrO and NO2. This can hardly be referred to as monitoring the ground for sinks.

P. 2, l. 16-18: Please clarify this statement. Kritten et al. (2014) describes studies on the photochemistry of NOx and Kreycy et al (2013) on BrOx. Weidner et al. (2005) describe merely an instrument and its performance. Kritten et al. (2010) present also mainly the technique and some diurnal variation of NO2. I quick search didn't show anything on trends. Do you maybe mean diurnal trends?

P. 2, l. 20-21: Baidar et al. (2013) describe measurements from a twin otter aircraft not HIAPER.

P. 2, l. 26: Meaningful instead of tractable? You can always do an inversion, also when you point in the wrong direction.

P. 2, l. 26: 'fed by' poor choice of wording

P. 2, l. 28-29: The stabilising system doesn't give the attitude data, but the attitude system provides this information and relays it to the stabilising system.

P. 2, l. 30-31: Again, you can always do an inversion, but it might not yield meaningful results. Also, absorption is not observed, but spectra. You also want to assign a trace gas concentration to a location in the atmosphere and not the absorption.

P. 2, l. 34: Celestial refers to what exactly?

P. 2, l. 33 – p. 3, l. 1: What you are describing constrains foremost the radiative transfer simulations. The inversion is constrained by an a priori. Please elaborate.

P. 3, l. 2: What gases if not O4? Please explain.

P. 3, l. 4: 'absorption strength' is rather vague.

P. 3, l. 6: Please add 'for airborne applications' after 'constraining the radiative transfer'

P. 3, l. 8-11. Following the previous sentence, you make it appear as if this is the O4's fault. But it should be described as ill-constrained by auxiliary parameters. I guess this is your 'second problem'. Please clarify.

P. 3, l. 11-13: Maybe I misunderstand something here. But if there are no aerosols, you cannot assign this lack to the wrong profile layer. Please elaborate.

P. 3, l. 18: 'Wrongly called Fraunhofer reference spectrum'!! I keep on coming across this phrase in publications. It is an 'in-term' which excludes newcomers to the field of DOAS by confusing them with wrong terminology. Please don't use this or elaborate that it has been used historically, but is not precise.

P. 3, l. 18-19. The Lambert-Beer law calls for a background spectrum.

P. 3, l. 21-22, l. 24, l. 25, l. 29, P. 8, l. 16: 'Fraunhofer spectrum' s.a.

P. 3, l. 26-27: Volkamer et al. use zenith spectra.

P. 3, l. 33-34: None of the theses can be accessed without at least a link.

P. 3, l. 34: What sets your current study apart from Stutz et al and Werner et al.? This information is given on P. 4, l. 14 only. Maybe rearrange.

P. 4, l. 1: s.a.: you're not measuring absorption.

P. 4, l. 2: 'convenient' poor choice of word.

P. 4, l. 10-11: Please add references for models.

P. 4, l. 11-13. 'convenient' poor choice of word. Sarcastically, I could state that it is indeed convenient that you validate what you put in as a priori information.

P. 4, l. 28: 'The' instead of 'Its'.

P. 5, l. 12: Spectrometers including detectors?

P. 5, l. 12: Maybe mention here already that the 6 different spectrometers are for 2 telescope geometries and 3 wavelength ranges.

P. 5, l. 31 and p. 6, l. 13: 'onto the lid' implies they are outside that container. Do you mean 'on the underside'?

P. 6, l. 1: I think this is the first time the detectors are mentioned.

P. 6, l. 10: 'subsequently' and 'prior to the flight' is not sufficiently explained.

P. 6, l. 24: 'subset of parameters' has a slightly negative connotation. I ask myself here, what you possibly might have neglected to characterize and why.

P. 6, l. 26: 'fields of view'

P. 7, l. 4, l. 16, and l. 17: play is more commonly referred to as backlash of a gear.

P. 7, l. 21: 'arguably' poor choice of word.

P. 7, l. 33: is below 0.2 deg acceptable?

P. 8, l. 4: The term dSCDs is mainly used for scattered sunlight DOAS and not for active DOAS techniques. So your statement is not fully correct.

P. 8, l. 8-9: And what do you do for the IR?

P. 10, l. 5: 'can be determined'?

P. 10, l. 15: 'their' refers to what?

P. 10, l. 22: RT was used in the previous section.

P. 10, l. 25: And what is the non-standard run?

P. 10, l. 26, p. 11, l. 3: Please rephrase 'fed'.

P. 10, l. 27: EA?

P. 10, l. 30: 'celestial'? 'et cetera'?

P. 11, l. 6: Reference for FAIRO?

P. 12, l. 22: 'in the lower troposphere'?

P. 13, l. 7: Reference to Table 2 doesn't help here if you don't state which are the target gases and which the scaling gases.

P. 13, l. 8-9: 'potential'?

P. 13, eq. 6: why don't you define a B_i here then? This is unnecessarily complicated. See above.

P. 14, l. 12: 'are obtained from Eq. (1)': I think you should mention that they are obtained from a DOAS fit and then Eq. (1).

P. 14, l. 15-24. This is very complicated. You need to read the text, the caption and the figure at the same time to get all the information to understand what is going on. Curtain is a confusing term, so are the random factors. I would need to get a calculator out to assign a specific colour to a number. Why did the authors not choose a non-fractional scaling factor for the units? Where does the uncertainty in alpha_R come from at this point and how is the uncertainty in the SCD ratio calculated? Please state here where the uncertainties will be described in the manuscript.

P. 14, l. 26: Please define 'compact relationship'

P. 14, l. 26-27: This basically only confirms that the remote sensing aircraft measurements are mainly sensitive to the concentrations at the flight altitude which was shown before (e.g., Baidar et al., 2013; Bruns et al., 2006). The citations I assume refer to the McArtim code? Again, I cannot access these without a link.

P. 15, l. 19-21: Also p and T will have horizontal and vertical gradients and those will depend on the aircraft altitude. The same is valid for using ozone as scaling gas. Please discuss.

P. 15, l. 22: Why do the authors not use proper error propagation here?

P. 15, l. 24-26: Please explain why you are using percentage values for the errors. These numbers won't be applicable for small SCDs.

P. 15, l. 29 – P. 16, l. 10: The supplementary material only shows results for UV and not visible as stated in the manuscript. I have problems understanding Figure 4 in the supplements considering the limited information provided: what are the frequency distributions? Is the altitude the aircraft altitude?

P. 16, l. 8: Then why do you use formaldehyde as a representative case here?

P. 16, l. 10-17: In the supplementary Figure 5a), is the in situ data filtered or smoothed?

P. 17, l. 16: 'validate' is a strong word in the context provided by this section.

P. 17, l. 20: 'agree reasonable well': That is impossible to assess from the figure.

P. 17, l. 20-22: Please don't use absorption here. You are referring to the different absorber concentration profiles. These two sentences are rather misleading as they are right now.

P. 18, l. 17: 'curtains' s.a.

P. 18, l. 27: This is quite an abrupt transition to BrO!

P. 18, l. 29: Isn't Figure 9 during flight section C?

P. 19, l. 9: That could still mean that both models are wrong.

P. 19, l. 20-21: Where do these detection limits come from?

P. 19, l. 21: Maybe replace 'degrade' by 'decrease' or 'is lost'.

P. 20, l. 7: Remove 'eventually'

P. 20, l. 9-10: Why do you introduce FT, PBL, and LMS abbreviations here?

P. 20, l. 10-11: How are they compatible? The previously mentioned studies Fitzenberger et al., and Prados-Roman et al. are both from the Arctic and they do show elevated levels of BrO in the FT. Please clarify your statement.

P. 20, l. 20: Maybe not name GLORIA at this point now otherwise I would ask for more explanations.

P. 20, l. 24-25: 'skylight radiances'? Is the instrument radiometrically calibrated?

P. 21, l. 9-11: See comment above, you basically get as output, what you provide as input. However, both models could still be wrong.

P. 21, l. 14: Where do 3.5 km and 15 km come from all of a sudden?

P. 21, l. 14: 'It can be argued': Please rephrase. 'major'?

Table 2: Formatting seems a bit messed up. Maybe add horizontal lines or larger gaps between the lines for the different trace gases.

Table 4: Is not referred to in the text.

Figure 2: Please add the concentrations to the figure or caption. 'Tob' in caption.

Supplementary Figure 7: and captions in the supplementary don't explain the red line in panel b.

Kritten et al. (2010) has wrong title. It's the AMTD title which was then renamed for the AMT version.

---

## Author Comment (AC1) · 19 Sep 2017

Hüneke et al., The novel HALO mini-DOAS instrument: Inferring trace gas concentrations from air-borne UV/visible limb spectroscopy under all skies using the scaling method

Answers to Reviewer 1

The reviewer's comments are in black, answers are in red.

The manuscript describes the mini-DOAS instrument that has already been operated in a number of science campaigns onboard the German High Altitude and Long-range research aircraft (HALO) together with the so-called "scaling method" to infer trace gas mixing ratios from the mini-DOAS limb measurements. The paper is generally well written and provides an important reference for the mini-DOAS instrument and in particular its data analysis. I recommend publication in AMT after consideration of the following – mostly minor – comments.

**Specific comments:**

As the instrument has already been flown on a number of science campaigns and this paper gives only a few selected examples, I suggest to include a table listing deployments of the mini-DOAS instrument on HALO campaigns so far.

*A new table listing all HALO campaigns in which the mini-DOAS instrument participated is included in the revised manuscript.*

p1.l3: please spell out HALO as "German High Altitude and Long-range research aircraft (HALO)" when used for the first time and move the URL reference from the abstract to the introduction section.

*The text is rephrased accordingly.*

p2.l1: sentence ends early

*The text is rephrased accordingly.*

p2.l6: better either give an earlier reference for DOAS as well, or explicitly indicate that Platt and Stutz (2008) is a recent review of DOAS and not the original reference

*The text is rephrased accordingly.*

p2.l22: "German GV" -> "German High Altitude and Long Range Research Aircraft (HALO), that is based on a Gulfstream G550 jet"

*The text is rephrased accordingly.*

p2.l35: what exactly is meant by "celestial" here?

*‚Celestial' is a standard expression in astronomy and it refers to the position of astronomical objects, here the sun and the earth.*

p3.l29: "to" -> "to be"

*The text is rephrased accordingly.*

p4.l11: what exactly does "latter" here refer to? Only CCMs or CTMs and CCMs?

*It refers to the incorporated model predictions. The text is rephrased accordingly.*

p5.l18: insert "elevation angles"

*The text is rephrased accordingly.*

p5.l20: why "skylight" and not simply "light"0? I know its picky, but in general there may be other light sources than just skylight.

*The text is rephrased accordingly.*

p6.l9: Can you explain briefly why surface temperatures are important here and not the much cooler upper tropospheric / lower stratospheric temperatures at cruise altitude?

*In this context the relevant time frame is the duration of the pre-flight preparations of the aircraft, while it is located on the apron. Access to the instrument is not possible and at large ambient T's some of the ice-water already melts affecting the instrument in-flight temperature on longer lasting sorties. The text is rephrased to make this clear.*

p7.l10: any reference to BAHAMAS ?

*No, unfortunately there is no proper reference for BAHAMAS available.*

p9.l1: "Polstracc" -> "POLSTRACC"

*The text is rephrased accordingly.*

P9.l14: "Acridicon" -> "ACRIDICON"

*The text is rephrased accordingly.*

p10.l27: What is "EA"? Elevation angle?

*EA is the elevation angle, as mentioned in section 2.4.*

p11.l6: "Fairo" -> "FAIRO"

*The text is rephrased accordingly.*

p11.l7: "for measuring" -> "for in-situ measuring"

*The text is rephrased accordingly.*

p11.l16: not sure if "all 13 flights" is still relevant here, as many more flights on more recent campaign have been performed with FAIRO

*The statement refers to the fact that the FAIRO instrument was first deployed on HALO during the ESMVal campaign.*

p12.l17: horizontal resolution is not well defined for a Lagrangian model. Please give more detail what this refers to.

*The horizontal resolution refers to the average distance between two neighboring air parcels in one model layer (see Vogel et al. (2015) and references therein and McKenna, Daniel S., et al. "A new Chemical Lagrangian Model of the Stratosphere (CLaMS) 1. Formulation of advection and mixing." Journal of Geophysical Research: Atmospheres 107.D16 (2002)).*

p12/13: EMAC: Maybe include a few sentences about EMAC's chemistry scheme – in contrast or comparison to what has been mentioned for CLaMS.

*In contrast to ClaMS, EMAC contains a very detailed tropospheric chemistry scheme. For the present EMAC model run, the submodel MECCA (Module Efficiently Calculating the Chemistry of the Atmosphere) is used to simulate the chemical kinetics, with the photochemical data taken from the JPL compilation (Sander et al., 2011) including recent updates (Jöckel et al., 2016). The details are presented by Jöckel et al. (2016), Section 3.5 on pages 1160/1161.*

*The text is added to the manuscript.*

p14.l12: Suggestion: Say again explicitly that SCD_X and SCD_P are measured by the mini-DOAS, the alpha are from a model and [P] is measured in-situ. (If that is what you are doing.)

*The text is rephrased accordingly.*

p20.l11: better be a bit more specific of where no elevated BrO concentrations are observed. I assume this statement does not refer to the Antarctic boundary layer?

*This refers to the Antarctic free troposphere. The text is rephrased accordingly.*

Caption Fig. 2: Suggestion: Include in the caption date, location and campaign of sample measurement.

*The text is rephrased accordingly.*

---

## Author Comment (AC2) · 19 Sep 2017

Hüneke et al., The novel HALO mini-DOAS instrument: Inferring trace gas concentrations from air-borne UV/visible limb spectroscopy under all skies using the scaling method

Answers to Reviewer 2

The reviewer's comments are in black, answers are in red.

The authors describe an airborne remote sensing instrument for UV/vis/near IR scattered sunlight measurements and the method for retrieving trace gas concentrations for the flight altitude, namely DOAS and the scaling method. Further the uncertainties are described to some extent, and some first application together with a comparison with auxiliary in situ measurements are presented.

I see two main problems with the manuscript and the scaling method: a) the uncertainty on the retrieved trace gas concentrations associated with the assumptions on the profile shape in the calculation of the scaling factor appears to be the dominant uncertainty and it is poorly quantified (partially hidden away in the supplementary and mentioned in section 4.2) and b) information on the radiative transfer in the atmosphere can only be obtained from layers where the scaling gas is actually present (This is hinted at in section 4.1). The authors claim that the scaling method is superior to optimal estimation. However, the advantage of optimal estimation is that it provides a formal framework for assessing the information content of a retrieved quantity. I believe the paper would be much stronger if these issues would be properly discussed upfront. Hence I recommend publication only after major revisions in sections 3.6, 3.7, 4, and 6.

*Our reactions to the reviewer's major concerns & comments:*

a) The uncertainty on the retrieved trace gas concentrations associated with the assumptions on the profile shape in the calculation of the scaling factor appears to be the dominant uncertainty and it is poorly quantified.

*The reviewer is correct in pointing out the uncertainty resulting from the assumed profile shapes. However, uncertainties of profile shapes taken from chemistry transport or chemistry climate models are hard to quantify. Here it should again be pointed out that when applying the scaling method only relative profile shapes matter but not absolute concentrations. Hence, in order to quantify the sensitivity of the scaling method on the actual profiles, the following sensitivity studies on the scaling factor were performed: (1) including a small scale variability derived from the O3 in-situ measurements, (2) including the uncertainty due to the vertical sampling error of the profile factor (see section 3.7.1), and (3) shifting the profile vertically in an arguably extreme manner to account for systematic errors in the adopted vertical dynamics (i.e. the adopted cooling/heating rate) in the models (see supplement, Figure 7). Furthermore, Stutz et al. (2017) carried out a range of sensitivity studies modifying trace gas profile shapes by altering concentrations overhead and below the aircraft (see their Table 5 and supplement). In fact, they arrived at uncertainties of the scaling factor similar to those obtained in our study.*

*Nevertheless, not all imaginable variations of the profile shapes can be covered in such sensitivity studies and in particular strong trace gas concentration gradients such as at the polar vortex edge pose problems for the method, as mentioned in the text (section 3.7.2). Besides such specific situations, one can reasonably assume that the inferred trace gas concentrations are conservative with respect to the model predicted profiles, i.e. possible deviations from the model predictions are underestimated. Mainly this conclusion is drawn from the observation that e.g. if the actual trace gas concentration was lower than the model prediction at sampling altitude (with all other altitudes the same), then the alpha factor for that trace gas would probably be overestimated, resulting in an overestimated trace gas concentration (compare Eq. 11).*

b)  Information on the radiative transfer in the atmosphere can only be obtained from layers where the scaling gas is actually present.

*The statement of the reviewer deserves some consideration, as in particular mentioned in section 4.1. In the best case, absorption by the scaling gas and the targeted gas occurs in similar altitude layers. It is shown in Figure 6 of the manuscript that by using an inappropriate scaling gas (in this case O4) the change in radiative properties (in this case the occurrence of a cloud layer at 4-8 km) can have a very strong influence on the retrieved NO2 concentrations above 8 km altitude. Comparably, when using O3 as a scaling gas, the influence of the low level clouds on retrieved NO2 concentrations is greatly diminished.*

The authors claim that the scaling method is superior to optimal estimation. However, the advantage of optimal estimation is that it provides a formal framework for assessing the information content of the retrieved quantity.

*The statement of the reviewer is at first glance reasonable, however only when assuming the forward model (to describe the observation) used in the optimal estimation reasonably describes the physical reality. While the RT can reasonably be approximated for limb measurements under clear (i.e. mostly Rayleigh scattering) skies, for cloudy skies any forward model lacks information on the spatial distribution, micro-physical and optical properties of the relevant scatterers, and their temporal variation over a single set of measurements. Noteworthy for ascent or descent of an aircraft or for a set of limb measurements at different elevation angles, a single set of measurements to obtain a profile may take tenths of minutes for which the RT has to be precisely known when applying optimal estimation.*
*Therefore, (many) assumptions need to made in the forward model describing the RT under clear skies, which propagate in an unquantifiable manner into the formal solution. In consequence, the information content formally returned by the optimal estimation algorithm is largely flawed, and not yet really quantified.*
*Further, as shown in previous studies (Stutz et al., 2017, Werner et al, 2017) the relevant RT is at least 2D (clear skies) and under cloudy skies 3D over domains of hundreds of kilometers in the limb direction for measurements in the upper troposphere and lower stratosphere.*
*In order to overcome these problems - applying optimal estimation for cloudy sky studies - the scaling method has been developed and intensive sensitivity studies have been performed for a wide range of conditions (Knecht, 2015, Stutz et al., 2017, Werner et al, 2017). Moreover, unlike stated by the reviewer the scaling method also offers information content of the measurements, here expressed in alpha factors. They describe the fraction of total measured absorption due to line of sight absorption (i.e. in field of view of the telescope).*

I believe the paper would be much stronger if these issues would be properly discussed upfront. Hence I recommend publication only after major revisions in sections 3.6, 3.7, 4, and 6.

*In consideration of the general and specific comments of the reviewer, the draft has been modified and explanations added.*

**General comments:**

P. 13-14: I'm not sure why the authors need such a lengthy mathematical description. In principle it is an intercept theorem: […] Intuitively, this is easier to understand.

*The reviewer is correct in proposing that the approach is similar to an intercept theorem, however only if the profile shapes of the involved trace gases were the same. The suggestion misses the point that by calculating alpha factors the fraction of the absorption collected from the atmosphere which is located within the instrument's line of sight is calculated. Hence, the sensitivity of the measurement to detect the target gas at flight altitude can be estimated, in some analogy with the averaging kernels obtained in the optimal estimation.*

P. 16, l. 28 – P. 17, l. 8: I would suggest using the uncertainty of the model/climatological profiles and do a proper error propagation to estimate the uncertainty. The NO2 mixing ratio is up to four times larger than the standard case at one point in the probed flight. Is that realistic?

*For a discussion of the uncertainty of the modelled profile shapes, see our answer to the reviewer's major comments and concerns above. Evidently large relative errors result when the target gas has small concentrations at flight level as compared to all atmospheric layers from which the absorption of the targeted gas is collected. Exactly this situation occurs for the measurements of NO2 within the polar vortex, and a change in profile shape at the vortex edge can therefore lead to strong deviations.*

P. 18, l. 6-9: This shouldn't be just discussed as differences between different model parametrisations. Basically what it says it that you need good prior knowledge in order to retrieve a meaningful concentration with the scaling method.

*The comparison shows that the prior knowledge (in this case the relative trace gas profile shapes) does have an impact on the retrieved concentrations, as is the case with any inversion method which uses prior knowledge. The point here is to show that while dynamical and chemical differences between models may render them more or less suitable for our task, the differences in the outcome are not unreasonable large to a degree that they heavily impact the result i.e., beyond the stated errors.*

P. 3, l. 33-34: The scaling method can hardly be described as novel method when the authors themselves already quote 6 reference for it.

*The scaling method is novel in the sense that it has only been used in a few studies so far (Stutz et al., 2017), but not for the interpretation of air-borne limb studies in the heavily aerosol and cloud-loaded troposphere.*

P.1, l. 4-5: Since the paper is about the scaling method, may be add a bit more explanation in the abstract what is actually done.

*The manuscript describes a novel air-borne DOAS instrument as well as the scaling method and its characterisation. Considering that an abstract should remain concise and short, the sentence „Here we report on the relevant instrumental details and the novel scaling method used to infer the mixing ratios of UV/vis absorbing trace gases from their absorption measured in limb geometry. The uncertainties of the scaling method are assessed in more detail than before for sample measurements of NO2 and BrO." may reflect well the contents of the manuscript.*

P. 21, l. 15-21: The a priori for optimal estimation is also a trace gas profile. Aerosol and cloud profiles are auxiliary information. The a priori for the scaling method is the profile ratio and aerosols and clouds are still auxiliary information. You have shown that those can be neglected. However, this is just how RT works for airborne measurements and hence this statement is valid for both techniques.

*The phrasing concerning „a priori" versus „parameters constraining radiative transfer" is corrected in the text. However, we strongly disagree with the reviewer's comment that 'aerosols and clouds can be neglected' in the forward modelling of spectroscopic*

*measurements when applying 'both techniques'. The reverse is true since any inversion tries to minimize the square distances of measured vs modelled parameter(s) (called cost function), and hence the model needs to reflect somehow the physical reality in order for any inversion to make sense (e.g. Rodgers, 2000, Tarantola, Inverse Problem Theory and Methods for Model Parameter Estimation, Society for Industrial and Applied Mathematics (SIAM), 2005). Evidently since our technique relies on the analysis of light, the forward (RT) model needs to account for all processes affecting the propagation of light, i.e. the presence of aerosol and clouds in the terrestrial atmosphere.*

*Since often remote sensing measurements in the UV/vis spectral range suffer from a lack of information in the spatial distribution of the aerosols and clouds and their optical properties, other means to constrain the inversion have been developed in the past, for example, by using the measured relative intensities (e.g. Prado-Roman et al., 2011), or the slant column densities of $O_4$ (e.g. Bruns et al., 2006), or any other suitable proxy.*

*In particular, it was recently shown (Stutz et al. (2017) and others) that $O_4$ is an inadequate constraint on the radiative transfer when inverting for high-altitude limb measurements using optimal estimation. Evidently the same is true when using $O_4$ as a constraint for the RT applying the scaling method (see our Figure 6). If there were other measured parameters available (being less sensitive to aerosol and clouds than $O_4$ or relative radiances) to constrain the radiative transfer in optimal estimation, we would probably agree with the reviewer's comment. However, we are not aware of any suitable atmospheric parameters of that kind.*

The authors poorly describe the need for the scaling method in the introduction. They list 3 main problems with airborne DOAS measurements on page 3. However, the first two problems are somewhat convoluted and boil down to the remote sensing technique being ill-constrained. The third problem is about the residual trace gas amount in the background measurement. This seems somewhat inflated in its description (also in section 3.2 where the applied technique is then described). The authors then don't make use of their own arguments to describe the scaling method for providing a different a priori information in comparison to what has been done before.

*In the introduction two major problems with air-borne DOAS measurements are addressed, i.e. (1) how to determine the absorption in the background (Fraunhofer) spectrum and (2) how to deal with largely unconstrained radiative transfer when aerosol and clouds are present. In order to bring more structure into the text we switched the paragraphs on page 2, line 30, and page 3, line 17, and rephrased them.*

The authors are not very precise in their referencing (see examples below). Please check carefully throughout the manuscript.

*Thank you for noting. We accordingly changed the citations at P. 2, l. 16-18.*

**Specific comments:**

P.1, l. 11-13: not a complete sentence

*We changed the text accordingly.*

P.2, l. 1-2: 'It's remoteness initiated…'. I feel the authors are overstating here. Another reason for aircraft measurements would be to study the full extent of the ozone hole and not only locally. Please clarify or add reference for original statement.

*We changed the text accordingly.*

P. 2, l. 8: 'aircraft-borne' sounds clunky.

*All occurrences of 'aircraft-borne' are replaced with 'air-borne'.*

P. 2, l. 11: How do you obtain information of 'photochemistry of pollutants' from column measurements? Please elaborate.

*Column measurements are often used to infer photochemical parameters, such as the life time of NO2 (e.g. Beirle et al., Megacity Emissions and Lifetimes of Nitrogen Oxides Probed from Space, Science, Vol. 333, Issue 6050, pp. 1737-1739 DOI: 10.1126/science.1207824, 2011.) Here, we added the example of HONO measurements by the CARIBIC instrument (Heue et al., CARIBIC DOAS observations of nitrous acid and formaldehyde in a large convective cloud, ACP, https://www.atmos-chem-phys.net/14/6621/2014/, DOI: 10.5194/acp-14-6621-2014, 2014).*

P. 2, l. 13-15: '… monitor the ground for sources and sinks…': Of the cited references only General et al. (2014) describes an anti-correlation between BrO and NO2. This can hardly be referred to as monitoring the ground for sinks.

*However, keeping the notation 'sinks' is justified here, given that there are many studies on sinks of trace gases inferred from remote sensing measurements. Here we just add one example: Beirle et al., Megacity Emissions and Lifetimes of Nitrogen Oxides Probed from Space, Science, Vol. 333, Issue 6050, pp. 1737-1739 DOI: 10.1126/science.1207824, 2011.*

P. 2, l. 16-18: Please clarify this statement. Kritten et al. (2014) describes studies on the photochemistry of NOx and Kreycy et al (2013) on BrOx. Weidner et al. (2005) describe merely an instrument and its performance. Kritten et al. (2010) present also mainly the technique and some diurnal variation of NO2. I quick search didn't show anything on trends. Do you maybe mean diurnal trends?

*Not really. Kreycy et al (2013) deals with the coupling of NO2 and BrO. The sentence is accordingly rephrased for a more accurate citation. Since the context of trends would need further elaboration, this part is dropped from the sentence.*

P. 2, l. 20-21: Baidar et al. (2013) describe measurements from a twin otter aircraft not HIAPER.

*We changed the text accordingly.*

P. 2, l. 26: Meaningful instead of tractable? You can always do an inversion, also when you point in the wrong direction.

*We changed the text accordingly.*

P. 2, l. 26: 'fed by' poor choice of wording

*We changed the text accordingly.*

P. 2, l. 28-29: The stabilising system doesn't give the attitude data, but the attitude system provides this information and relays it to the stabilising system.

*We changed the text accordingly.*

P. 2, l. 30-31: Again, you can always do an inversion, but it might not yield meaningful results. Also, absorption is not observed, but spectra. You also want to assign a trace gas concentration to a location in the atmosphere and not the absorption.

*We changed the text accordingly.*

P. 2, l. 34: Celestial refers to what exactly?

*The word celestial is a standard expression in astronomy and it refers to the position of astronomical objects, here the sun and the earth.*

P. 2, l. 33 – p. 3, l. 1: What you are describing constrains foremost the radiative transfer simulations. The inversion is constrained by an a priori. Please elaborate.

*The constrains/boundary conditions and assumptions are described foremost because they largely determine the actual RT. We changed the text accordingly.*

P. 3, l. 2: What gases if not O4? Please explain.

*In particular, here those gases are meant which have a sizeable effect on the radiative transfer in the present context (i.e. are optically thick, or tau >1 ) .*

P. 3, l. 4: 'absorption strength' is rather vague.

*We changed the text accordingly by using the notation ‚slant column density'.*

P. 3, l. 6: Please add 'for airborne applications' after 'constraining the radiative transfer'

*The suggested phrase is added at the beginning of the sentence.*

P. 3, l. 8-11. Following the previous sentence, you make it appear as if this is the O4's fault. But it should be described as ill-constrained by auxiliary parameters. I guess this is your 'second problem'. Please clarify.

*The text merely describes the sensitivities when using O4 as a parameter to constrain radiative transfer, and points out the insufficiency of using it as RT proxy for limb measurement from high-flying, fast-moving aircraft platforms, which is exactly what the reviewer suggests. The paragraph is rephrased to make clear what is meant by the 'first' and 'second' problem.*

P. 3, l. 11-13: Maybe I misunderstand something here. But if there are no aerosols, you cannot assign this lack to the wrong profile layer. Please elaborate.

*If there were no aerosols and clouds in the atmosphere, then they would not need to be considered and the optimal estimation would work out just fine. However, if the scattering properties of the atmosphere changed during the measurements, the inferred O4 slant column density in limb would change too with changing cloud cover underneath even when flying at constant altitude. In consequence, an apparent aerosol layer located at higher altitudes would be retrieved.*

P. 3, l. 18: 'Wrongly called Fraunhofer reference spectrum'!! I keep on coming across this phrase in publications. It is an 'in-term' which excludes newcomers to the field of DOAS by confusing them with wrong terminology. Please don't use this or elaborate that it has been used historically, but is not precise.

*The sentence is rephrased accordingly and the notation ‚background spectrum' is used throughout the text.*

P. 3, l. 18-19. The Lambert-Beer law calls for a background spectrum.

*We changed the text accordingly by rephrasing the sentence.*

P. 3, l. 21-22, l. 24, l. 25, l. 29, P. 8, l. 16: 'Fraunhofer spectrum' s.a.

*The sentence is rephrased accordingly.*

P. 3, l. 26-27: Volkamer et al. use zenith spectra.

*The sentence is rephrased accordingly.*

P. 3, l. 33-34: None of the theses can be accessed without at least a link.

*Appropriate links are added in the bibliography.*

P. 3, l. 34: What sets your current study apart from Stutz et al and Werner et al.? This information is given on P. 4, l. 14 only. Maybe rearrange.

*We changed the text in the abstract accordingly.*

P. 4, l. 1: s.a.: you're not measuring absorption.

*We changed the text accordingly.*

P. 4, l. 2: 'convenient' poor choice of word.

*We changed the text accordingly.*

P. 4, l. 10-11: Please add references for models.

*A reference to section 3.5 is added, where the models are described.*

P. 4, l. 11-13. 'convenient' poor choice of word. Sarcastically, I could state that it is indeed convenient that you validate what you put in as a priori information.

*We accordingly changed the text to make more clear what is meant. In fact, in the past we collected many examples where the CTM predictions and our measurements as well as the complementary measurements do not agree at all (e.g. on NO/NO2 ratio, or the presence of tropospheric BrO, IO, and CH2O ….), hence any incorrect 'a priori' information drawn from the model is not carried over to the final result. Another good example on the power of the scaling method and its comparably weak reliance on any 'a priori' information can be found in the publication of Ye et al., (2016) (Rapid cycling of reactive nitrogen in the marine boundary layer, Nature, 532, 489–491, (28 April 2016).*

P. 4, l. 28: 'The' instead of 'Its'.

*We changed the text accordingly.*

P. 5, l. 12: Spectrometers including detectors?

*Modern optical spectrometers usually contain detectors.*

P. 5, l. 12: May be mention here already that the 6 different spectrometers are for 2 telescope geometries and 3 wavelength ranges.

*The first sentence of section 2.1 is rephrased accordingly.*

P. 5, l. 31 and p. 6, l. 13: 'onto the lid' implies they are outside that container. Do you mean 'on the underside'?

*In the first instance, it is the lower side, in the second, it is the top of the container. We changed the text accordingly.*

P. 6, l. 1: I think this is the first time the detectors are mentioned.

*The detectors are now explicitly mentioned in the beginning of section 2.2.*

P. 6, l. 10: 'subsequently' and 'prior to the flight' is not sufficiently explained.

*We changed the text accordingly.*

P. 6, l. 24: 'subset of parameters' has a slightly negative connotation. I ask myself here, what you possibly might have neglected to characterize and why.

*The notation "subset" refers to those parameters employed in the DOAS retrieval. We changed the text accordingly.*

P. 6, l. 26: 'fields of view'

*We changed the text accordingly.*

P. 7, l. 4, l. 16, and l. 17: play is more commonly referred to as backlash of a gear.

*The sentences have been rephrased accordingly in the text.*

P. 7, l. 21: 'arguably' poor choice of word.

*The sentence has been rephrased to reflect that the given numbers are typical examples.*

P. 7, l. 33: is below 0.2 deg acceptable?

*Certainly yes, since due to (multiple) scattering the RT in UV/vis spectral range already degrades the effective field of view to larger angles. This can for example be learned when inspecting the width of averaging kernels for similarly stabilized instruments (Baidar at al., 2013), or from the intensive RT exercises we and others performed in the recent past (cf. Raecke, 2013, http://www.iup.uni-heidelberg.de/institut/forschung/groups/atmosphere/stratosphere/publications/pdf/MScThesis_Rasmus_Raecke.pdf, Knecht, 2015 http://www.iup.uni-heidelberg.de/institut/forschung/groups/atmosphere/stratosphere/publications/pdf/MA_thesis_Matthias_final.pdf )*

P. 8, l. 4: The term dSCDs is mainly used for scattered sunlight DOAS and not for active DOAS techniques. So your statement is not fully correct.

*The sentence is rephrased accordingly.*

P. 8, l. 8-9: And what do you do for the IR?

*Here are two examples for studies carried out using nearIR measurements: 1. Wolf et al., Potential of remote sensing of cirrus optical thickness by airborne spectral radiance measurements at different sideward viewing angles, Atmos. Chem. Phys., 17, 4283-4303, doi:10.5194/acp-17-4283-2017, 2017. And 2. PhD thesis of Scalone, Retrieval of Cirrus Optical Properties in the near-IR spectral range within the NASA ATTREX Project, PhD Thesis, University of Heidelberg, Germany, 2017 (doi:10.11588/heidok.00023004, http://www.ub.uni-heidelberg.de/archiv/23004).*

P. 10, l. 5: 'can be determined'?

*We changed the text accordingly.*

P. 10, l. 15: 'their' refers to what?

*The word 'their' refers to the measured spectra of the flights where optimal estimation cannot be applied. A clarification is added to the text.*

P. 10, l. 22: RT was used in the previous section.

*We changed the text accordingly.*

P. 10, l. 25: And what is the non-standard run?

*The sentence is rephrased and for all simulations the settings are mentioned in the text (from line 25 on).*

P. 10, l. 26, p. 11, l. 3: Please rephrase 'fed'.

*We changed the text accordingly.*

P. 10, l. 27: EA?

*EA is the elevation angle, as mentioned in section 2.4.*

P. 10, l. 30: 'celestial'? 'et cetera'?

*The word celestial is standard in astronomy and hence used for the position of the sun and earth. We changed the text accordingly.*

P. 11, l. 6: Reference for FAIRO?

*The reference for FAIRO has been moved to the second sentence of the paragraph.*

P. 12, l. 22: 'in the lower troposphere'?

*We changed the text accordingly.*

P. 13, l. 7: Reference to Table 2 doesn't help here if you don't state which are the target gases and which the scaling gases.

*We changed the text accordingly (now Table 3).*

P. 13, l. 8-9: 'potential'?

*We removed potential from the text.*

P. 13, eq. 6: why don't you define a B_i here then? This is unnecessarily complicated. See above.

*See answer to general comments above.*

P. 14, l. 12: 'are obtained from Eq. (1)': I think you should mention that they are obtained from a DOAS fit and then Eq. (1).

*We changed the text accordingly.*

P. 14, l. 15-24. This is very complicated. You need to read the text, the caption and the figure at the same time to get all the information to understand what is going on. Curtain is a confusing term, so are the random factors. I would need to get a calculator out to assign a specific colour to a number. Why did the authors not choose a non-fractional scaling factor for the units? Where does the uncertainty in alpha_R come from at this point and how is the uncertainty in the SCD ratio calculated? Please state here where the uncertainties will be described in the manuscript.

*In present atmospheric science the notion curtain is a widely used notion for a (CTM) simulated (atmospheric) parameter as a function of time/horizontal coordinate and altitude. There is nothing random about it. However, we include a definition of the notion 'curtain' upon its first appearance in the text (page 11, line 4). Nice examples of curtains of potential vorticity and trace gases can also be found in recent publications of Jurkat et al., (2017) Figure 1 (Depletion of chlorine and nitrogen oxide reservoir species in the 2 lower Antarctic polar vortex measured from aircraft, Geophys. Res. Lett., 44, doi:10.1002/2017GL073270, 2017), Werner et al., (2017) (Figure 2). The fractional numbers just appear in order to project all gases onto a common colour scale. In fact, the application of scaling method does not even require consideration of the individual multiplier because for a specific gas a common multiplier cancels out (see eq. 4 and 5). The overall uncertainty is calculated via equation (13). SCD uncertainties are discussed on page 15, line 23 and uncertainties of the alpha factors are discussed on page 15, from line 27 on. A sentence has been added to point the reader to the respective section.*

P. 14, l. 26: Please define 'compact relationship'

*Figure 4 itself provides a good impression of 'a fairly compact relationship' (cited from the text) as compared to data which do not show any correlation.*

P. 14, l. 26-27: This basically only confirms that the remote sensing aircraft measurements are mainly sensitive to the concentrations at the flight altitude which was shown before (e.g., Baidar et al., 2013; Bruns et al., 2006). The citations I assume refer to the McArtim code? Again, I cannot access these without a link.

*Sorry. We added the link to the bibliography. In fact, it was recently shown by Stutz et al., (2017) (Figure 9). However, for a newly deployed instrument it appears prudent to show that it is indeed behaving the way it is supposed to. The sentence has been rephrased to reflect this.*

P. 15, l. 19-21: Also p and T will have horizontal and vertical gradients and those will depend on the aircraft altitude. The same is valid for using ozone as scaling gas. Please discuss.

*The basic idea of the method presumes that the in-situ measured concentration of the scaling gas (be it O3 concentration or O4 calculated from p and T) is representative of the average concentration of that gas in the atmospheric volume sampled by the instrument's telescopes. The alpha factor ratios are calculated for this volume. Hence, small scale variabilities in the concentration of the scaling gas can lead to an error, which is incorporated in the error estimate for the alpha factor ratio (see Table 5). Larger horizontal gradients in trace gas concentrations, e.g. when flying through the polar vortex edge, need to be carefully considered when interpreting the derived trace gas concentrations. We accordingly added a remark in section 3.7.2. The error due to the vertical gradient of the scaling gas is considered (1) by looking at the vertical sampling uncertainty of the telescopes (section 3.7.1, paragraph (c)) and (2) estimated by shifting trace gas profiles vertically to correct for model errors in vertical ascent/descent rates.*

P. 15, l. 22: Why do the authors not use proper error propagation here?

*As written and indicated in text, a Gaussian error propagation is applied.*

P. 15, l. 24-26: Please explain why you are using percentage values for the errors. These numbers won't be applicable for small SCDs.

*Uncertainties in percentages are given in order to make them comparable to the other factors influencing the overall error (compare Table 5). "Small" SCDs do not occur, since the reference-SCDs of limb background spectra measured in the troposphere and lowermost stratosphere always include at least the column of stratospheric O3 and NO2, e.g, NO2 reference-SCDs are $5*10^{15}$ … $1*10^{16}$ molec./cm^2, O3 reference-SCDs are approx. $1*10^{19}$ molec/cm$^2$, and O4 reference-SCDs are approx. $3*10^{43}$ molec$^2$/cm$^5$.*

P. 15, l. 29 – P. 16, l. 10: The supplementary material only shows results for UV and not visible as stated in the manuscript. I have problems understanding Figure 4 in the supplements considering the limited information provided: what are the frequency distributions? Is the altitude the aircraft altitude?

*The RT simulations were carried out in the UV (343 nm) as well as in the visible (477 nm). For details see Knecht (2015). The caption of Figure 4 in the supplement is rephrased to make clear what is meant.*

P. 16, l. 8: Then why do you use formaldehyde as a representative case here?

*The calculations were performed in support of the interpretation of the measurement taken within the framework of the ACRIDICON campaign (Wendisch et al., 2016, The ACRIDICON-CHUVA campaign to study tropical deep convective clouds and precipitation using the new German research aircraft HALO, Bull. Am. Meteorol. Soc., 97, 10, 1885-1908, doi:10.1175/BAMS-D-14-00255, 2016.) which took place in the Amazon basin in fall 2014. There, strong convection lead to cumulus and cumulonimbus clouds which are largely varied in vertical and horizontal extent, providing an extreme test case to determine the sensitivity of the method as a function of cloud cover .*

P. 16, l. 10-17: In the supplementary Figure 5a), is the in situ data filtered or smoothed?

*The in-situ data are smoothed in an interval ranging from 100 s before to 100 s after each spectroscopic measurement. A detailed description of the smoothing is added to the text of the supplement.*

P. 17, l. 16: 'validate' is a strong word in the context provided by this section.

*We changed the text accordingly.*

P. 17, l. 20: 'agree reasonable well': That is impossible to assess from the figure.

*Clarification is added to the text (differences are below 35 ppt).*

P. 17, l. 20-22: Please don't use absorption here. You are referring to the different absorber concentration profiles. These two sentences are rather misleading as they are right now.

*We exchanged the word 'absorption' with the word 'concentration' and rephrased the sentences.*

P. 18, l. 17: 'curtains' s.a.

*See above.*

P. 18, l. 27: This is quite an abrupt transition to BrO!

*The beginning of the subsection is rephrased accordingly.*

P. 18, l. 29: Isn't Figure 9 during flight section C?

*We changed the text accordingly.*

P. 19, l. 9: That could still mean that both models are wrong.

*For a discussion of model uncertainties see comment above.*

P. 19, l. 20-21: Where do these detection limits come from?

*The detection limits are estimated based on the uncertainties at very low mixing ratios. These are indicated in Figure 9, panels c and e. For [NO2] = 10 ppt and  [BrO] =2 ppt respectively, the inferred mixing ratios are at least two times larger than the uncertainty.*

P. 19, l. 21: Maybe replace 'degrade' by 'decrease' or 'is lost'.

*We changed the text accordingly.*

P. 20, l. 7: Remove 'eventually'

*We changed the text accordingly.*

P. 20, l. 9-10: Why do you introduce FT, PBL, and LMS abbreviations here?

*We changed the text accordingly.*

P. 20, l. 10-11: How are they compatible? The previously mentioned studies Fitzenberger et al., and Prados-Roman et al. are both from the Arctic and they do show elevated levels of BrO in the FT. Please clarify your statement.

*For the present DOAS retrieval we estimate our detection limit for BrO is 2 ppt (see comment above) and do not derive elevated BrO in the FT. BrO mixing ratios larger than 2 ppt were not shown either by Fitzenberger et al. or Prados-Roman et al. in the FT above 3.5 km (the minimum altitude of the measurements reported here).*

P. 20, l. 20: Maybe not name GLORIA at this point now otherwise I would ask for more explanations.

*The sentence is rephrased accordingly in the text.*

P. 20, l. 24-25: 'skylight radiances'? Is the instrument radiometrically calibrated?

*The instrument itself is not radiometrically calibrated, as mentioned in section 3.3. However, it can be cross-calibrated by comparison with a radiometrically calibrated instrument (see Wolf et al., 2017, their Figure 8).*

P. 21, l. 9-11: See comment above, you basically get as output, what you provide as input. However, both models could still be wrong.

*For the model uncertainties see the discussion above. Here it is necessary to recall that the result of the scaling method is not determined by the absolute concentrations (as predicted by the CTMs) but only by the relative profile shapes of the involved gases.*

P. 21, l. 14: Where do 3.5 km and 15 km come from all of a sudden?

*The phrase is dropped because it is not essential here. For altitudes covered by the present flight see Figure 3.*

P. 21, l. 14: 'It can be argued': Please rephrase. 'major'?

*We changed the text accordingly.*

Table 2: Formatting seems a bit messed up. May be add horizontal lines or larger gaps between the lines for the different trace gases.

*Horizontal lines are added for clarification (now Table 3).*

Table 4: Is not referred to in the text.

*Table 4 (now Table 5) is referred to in the beginning of section 3.7.1.*

Figure 2: Please add the concentrations to the figure or caption. 'Tob' in caption.

*The slant column densities of the targeted trace gases are added and the text corrected for spelling.*

Supplementary Figure 7: and captions in the supplementary don't explain the red line in panel b.

*The reference to the red line is added to the caption.*

Kritten et al. (2010) has wrong title. It's the AMTD title which was then renamed for the AMT version.

*The error in the bibliography is corrected.*